# 1 EMDNA: Ensemble Meteorological Dataset for North America

Guoqiang Tang[1,2], Martyn P. Clark[1,2], Simon Michael Papalexiou[2,3], Andrew J. Newman[4], Andrew
W. Wood[4], Dominique Brunet[5], Paul H. Whitfield[1,2]
[1]University of Saskatchewan Coldwater Lab, Canmore, Alberta, Canada
[2]Centre for Hydrology, University of Saskatchewan, Saskatoon, Saskatchewan, Canada
[3]Department of Civil, Geological and Environmental Engineering, University of Saskatchewan, Saskatchewan,
Canada
[4]National Center for Atmospheric Research, Boulder, Colorado
[5]Meteorological Research Division, Environment and Climate Change Canada, Toronto, Ontario, Canada
**Abstract:** Probabilistic methods are very useful to estimate the spatial variability in meteorological conditions (e.g.,
spatial patterns of precipitation and temperature across large domains). In ensemble probabilistic methods, "equally
plausible" ensemble members are used to approximate the probability distribution, hence uncertainty, of a spatially
distributed meteorological variable conditioned on the available information. The ensemble can be used to evaluate
the impact of the uncertainties in a myriad of applications. This study develops the Ensemble Meteorological Dataset
for North America (EMDNA). EMDNA has 100 members with daily precipitation amount, mean daily temperature,
and daily temperature range at 0.1° spatial resolution from 1979 to 2018, derived from a fusion of station observations
and reanalysis model outputs. The station data used in EMDNA are from a serially complete dataset for North America
(SCDNA) that fills gaps in precipitation and temperature measurements using multiple strategies. Outputs from three
reanalysis products are regridded, corrected, and merged using the Bayesian Model Averaging. Optimal Interpolation
(OI) is used to merge station- and reanalysis-based estimates. EMDNA estimates are generated based on OI estimates
and spatiotemporally correlated random fields. Evaluation results show that (1) the merged reanalysis estimates
outperform raw reanalysis estimates, particularly in high latitudes and mountainous regions; (2) the OI estimates are
more accurate than the reanalysis and station-based regression estimates, with the most notable improvement for
precipitation occurring in sparsely gauged regions; and (3) EMDNA estimates exhibit good performance according to
the diagrams and metrics used for probabilistic evaluation. We also discuss the limitations of the current framework
and highlight that persistent efforts are needed to further develop probabilistic methods and ensemble datasets. Overall,
EMDNA is expected to be useful for hydrological and meteorological applications in North America. The whole
dataset and a teaser dataset (a small subset of EMDNA for easy download and preview) are available at
https://doi.org/10.20383/101.0275 (Tang et al., 2020a).



## 1. Introduction

Precipitation and temperature data are fundamental inputs for a wide variety of geoscientific and operational applications benefitting society (Eischeid et al., 2000; Trenberth et al., 2003; Wu et al., 2014; Yin et al., 2018). Accurately estimating spatial meteorological fields is still challenging despite the availability of many measurement approaches (e.g., meteorological stations, weather radars, and satellite sensors) and atmospheric models (Kirstetter et al., 2015; Sun et al., 2018; Hu et al., 2019; Newman et al., 2019a). There is consequently substantial uncertainty in analyses of spatially distributed meteorological variables.

The uncertainty in spatial meteorological estimates depends on both the measurements available and the climate of the region of study. Whilst meteorological stations provide the most reliable observations at the point scale, spatial meteorological estimates based on station data can be degraded by the sparsity of station networks in remote regions and by measurement errors caused by factors such as evaporation/wetting loss and under-catch of precipitation (Sevruk, 1984; Goodison et al., 1998; Nešpor and Sevruk, 1999; Yang et al., 2005; Scaff et al., 2015; Kochendorfer et al., 2018). Interpolating station data to a regular grid can introduce additional uncertainties due to factors such as method choices and topographic variations. The accuracy of precipitation estimated from ground radars is affected by factors such as beam blockage, signal attenuation, ground clutter, and uncertainties in the reflectivity-rainfall relationships (Dinku et al., 2002; Kirstetter et al., 2015). Moreover, the spatial and temporal coverage of ground radars is limited to large populated areas in most regions of the world. Satellite sensors provide quasi-global estimates of meteorological variables, but their utility can be limited by short sampling periods with insufficient coverage and return frequency, indirect measurements, imperfect retrieval algorithms, and instrument limitations (Adler et al., 2017; Tang et al., 2016, 2020b). Reanalysis models, which provide long-term global simulations, also contain biases and uncertainties caused by the imperfect model representations of physical processes, observational constraints, model resolution, and model parameterization (Donat et al., 2014; Parker, 2016).

In recent years, numerous deterministic gridded precipitation and temperature datasets based on observed or simulated data from single or multiple sources have become available to the public (Maurer et al., 2002; Huffman et al., 2007; Mahfouf et al., 2007; Daly et al., 2008; Di Luzio et al., 2008; Haylock et al., 2008; Livneh et al., 2013; Weedon et al., 2014; Fick and Hijmans, 2017; Beck et al., 2019; Ma et al., 2020; Harris et al., 2020). Since the uncertainties vary in space and time, deterministic products do not always agree with each other (Donat et al., 2014; Henn et al., 2018; Sun et al., 2018; Newman et al., 2019a; Tang et al., 2020b). The uncertainties can be propagated to applications such as hydrological modeling and climate analysis (Clark et al., 2006; Hong et al., 2006; Slater and Clark, 2006; Mears et al., 2011; Rodell et al., 2015; Aalto et al., 2016). Proper understanding of the uncertainties can benefit the objective application of meteorological analyses and further improve existing products, yet few gridded datasets provide such uncertainty estimates (Cornes et al., 2018; Frei and Isotta, 2019).

Probabilistic datasets now can provide alternatives to deterministic datasets for quantitative precipitation and temperature estimation and have advantages in estimating uncertainties and representing extremes (Kirstetter et al.,





2015; Mendoza et al., 2017; Frei and Isotta, 2019). Recently, several ensemble meteorological datasets have become
available. For example, Morice et al. (2012) develop the observation-based HadCRUT4 global temperature datasets
with 100 members. Caillouet et al. (2019) develop the Spatially COherent Probabilistic Extended Climate dataset
(SCOPE Climate) with 25 members in France. Newman et al. (2015, 2019b, 2020) continually extend the probabilistic
estimation methodology proposed by Clark and Slater (2006), and produce ensemble precipitation and temperature
datasets in the contiguous USA (CONUS), the Hawaii Islands, and Alaska and Yukon, respectively. Moreover, several
widely used deterministic datasets now have ensemble versions in view of the advantages of probabilistic estimates.
Cornes et al. (2018) developed the ensemble version (100 members) of the Haylock et al. (2008) Europe-wide E-OBS
temperature and precipitation datasets. Khedhaouiria et al. (2020) developed the experimental High-Resolution
Ensemble Precipitation Analysis (HREPA) for Canada and the northern part of the CONUS with 24 members, which
can be regarded as an experimental ensemble version of the Canadian Precipitation Analysis (CaPA; Mahfouf et al.,
2007; Fortin et al., 2015).
Our objective is to develop an Ensemble Meteorological Dataset for North America (EMDNA) from 1979 to 2018.
To improve the quality of estimates in sparsely gauged regions, station data and reanalysis outputs are merged to
generate gridded precipitation and temperature estimates. Then, ensemble estimates are produced using the
probabilistic method described by Clark and Slater (2006) and Newman et al. (2015, 2019b, 2020). EMDNA has 100
members and contains daily precipitation amount, mean daily temperature (Tmean), and daily temperature range
(Trange) at 0.1° spatial resolution. Minimum and maximum temperature can be calculated from Tmean and Trange.
It is expected that the EMDNA will be useful for a variety of applications in North America.
## 2.  Datasets
Station observations often have missing values and short record lengths (Kemp et al., 1983). This study uses station
precipitation and minimum/maximum temperature data from the Serially Complete Dataset for North America (Tang
et al., 2020c), which is open-access on Zenodo (https://doi.org/10.5281/zenodo.3735533; Access Date: July 25, 2020).
Tmean and Trange are calculated from minimum and maximum temperature data. In SCDNA, raw measurements
undergo strict quality control checks, and data gaps are filled by combining estimates from multiple strategies.
SCDNA covers the period from 1979 to 2018 and has 24615 precipitation stations and 19579 temperature stations.
Station-based gridded meteorological estimates usually rely on a certain number of neighboring stations surrounding
the target location. For most regions in CONUS, the search radius to find 20 or 30 neighboring stations (lower and
upper limits for station-based gridded estimates in Sect. 3.1) is smaller than 100 km (Fig. 1). For the regions northern
to 50°N or southern to 20°N, however, the search radius is much larger and even exceeds 1000 km in the Arctic
Archipelago. The sparse station network at higher latitudes motivates our decision to optimally combine station data
with reanalysis products.
The reanalysis products used in this study include the fifth generation of European Centre for Medium-Range Weather
Forecasts (ECMWF) atmospheric reanalyses of the global climate (ERA5; Hersbach et al., 2020), the Modern-Era
Retrospective analysis for Research and Applications, Version 2 (MERRA-2; Gelaro et al., 2017), and the Japanese
55-year Reanalysis (JRA-55; Kobayashi et al., 2015). The spatial resolutions of ERA5, MERRA-2, and JRA-55 are
0.25°×0.25°, 0.5°×0.625°, and ~55 km, respectively. Their start years are 1979, 1980, and 1958, respectively.
Therefore, only ERA5 and JRA-55 are used for 1979 throughout this study. Although reanalysis models assimilate
observations from various sources, they differ with station measurements in many aspects (Parker, 2016) and often
contain large uncertainties as shown by assessment and multi-source merging studies (e.g., Donat et al., 2014; Lader
et al., 2016; Beck et al., 2017, 2019; Tang et al., 2020b). Thereby, the possible dependence between reanalysis
estimates and station data is not considered when merging them in this study.
The elevation data are sourced from the 3 arc-second resolution Multi-Error-Removed Improved-Terrain digital
elevation model (MERIT DEM; Yamazaki et al., 2017).

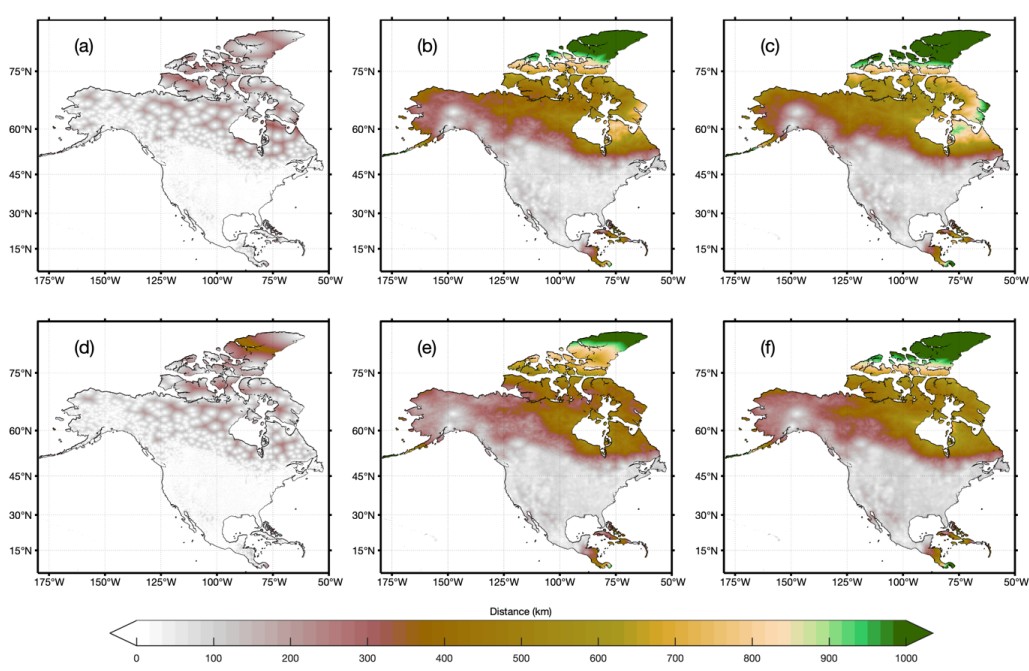


Figure 1. The color of each 0.1° grid indicates the radial radius to find (a) one, (b) 20, and (c) 30 neighboring stations
for precipitation (a-c) and temperature (d-f).





## 3. Methodology

The estimate of a variable at a specific location and time step can be regarded as a random value following a probability distribution. The probability density functions (PDFs) of variables such as the Tmean and Trange, can be approximated using the normal distribution. Their value $x$ for the target location and time step is expressed as:

$$x \sim N(\mu, \sigma^2) \tag{1}$$

where $\mu$ is the mean value and $\sigma$ is the standard deviation. The probabilistic estimation of Tmean and Trange can be realized by sampling from this distribution. In a spatial meteorological dataset, the distribution parameters vary with space and time, and the variability is related to the nature of variables and gridding (interpolation) methods. The performance of gridding methods is critical because accurate estimation of $\mu$ can reduce systematic bias and smaller $\sigma$ means narrower spread.

Precipitation is different from Tmean and Trange because it can be intermittent from local to synoptic scales and its distribution is both highly skewed and bounded at zero. Following Papalexiou (2018) and Newman et al. (2019b), the cumulative density function (CDF) of precipitation can be expressed as below:

$$F_X(x) = (1 - p_0)F_{X|X>0}(x) + p_0, \ for \ x \geq 0 \tag{2}$$

where $F_X(x)$ is the CDF for $x \geq 0$, $F_{X|X>0}(x)$ is the CDF for $x > 0$, and $p_0$ is the probability of zero precipitation. The probability of precipitation (PoP) is $1 - p_0$. The CDF $F_{X|X>0}(x)$ is often approximated using the normal distribution after applying suitable transformation functions to observed precipitation. Clark and Slater (2006) perform the normal quantile transformation using an empirical CDF from station observations. Newman et al. (2015) apply a power-law transformation. Newman et al. (2019b) adopts the Box-Cox transformation, that is,

$$x' = \frac{x^\lambda - 1}{\lambda} \tag{3}$$

where $\lambda$ is set to 1/3 following Newman et al. (2019b) and Fortin et al. (2015). Eq. (1) applies to $x'$, enabling the probabilistic estimation of precipitation. Unlike Newman et al. (2019b) that uses transformed precipitation throughout the production, this study only uses Box-Cox transformation when the assumption of normality is necessary (Sect. 3.2.4 and 3.3) to reduce the error introduced by the back transformation. The limitations and alternative choices of precipitation transformation are discussed in Sect. 5.2.

In summary, seven space- and time-varying parameters ($\mu$ and $\sigma$ for three variables and PoP) should be obtained to realize probabilistic estimation. Our method to develop probabilistic meteorological estimates is summarized in Fig. 2a. We apply four main steps to produce EMDNA: (1) station-based regression estimates (Sect. 3.1), (2) the regridding,

downscaling, bias correction and merging of three reanalysis products (Sect. 3.2), (3) optimal interpolation-based
merging of reanalysis and station-based regression outputs, and the bias correction of the resulting precipitation
estimates (Sect. 3.3), and (4) the production of probabilistic estimates in the form of spatial meteorological ensembles
(Sect. 3.4).

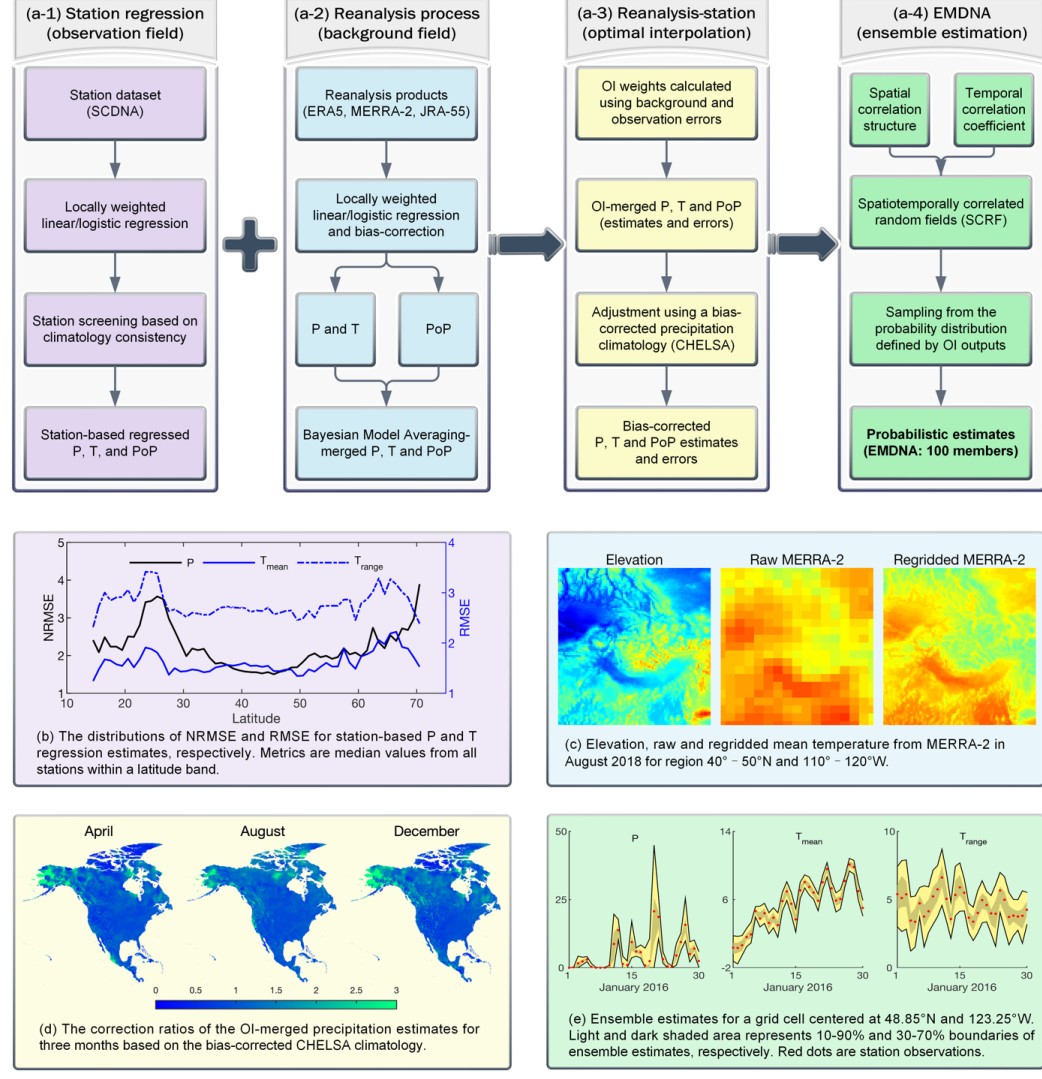


Figure 2. (a) The flowchart outlining the main steps for producing EMDNA. P represents precipitation and T represents
temperature. (b-e) demonstrate output examples from (a-1 to -4), respectively. (b) Latitudinal distribution of the root
mean square error (RMSE) for temperature and normalized RMSE (NRMSE) for precipitation (Sect. 3.1). (c) Example
showing the mean temperature of MERRA-2 before and after regridding (Sect. 3.2). (d) The correction ratios



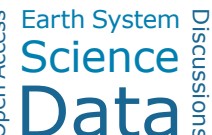

calculated using precipitation climatology from the bias-corrected CHELSA (Sect. 3.3). (e) Example of the ensemble-
based distributions of precipitation and temperature estimates from EMDNA (Sect. 3.4).

## 3.1 Regression estimates from station data

Clark and Slater (2006) and Newman et al. (2015, 2019b) use locally weighted linear regression and logistic regression
to obtain gridded precipitation and temperature estimates which are used as parameters in Eq. (1). However, for high-
latitude regions in North America where stations are scarce (Fig.1), such gridded estimates based only on station data
could contain large uncertainties (Fig. 2b) due to the long distances needed to assemble a sufficient sample of stations
to form the regressions. This study uses optimal interpolation (OI) to merge data from stations and reanalysis models.
In this section, we only obtain regression estimates and their errors at the locations of stations, which are used as inputs
to OI in Sect. 3.3.

### 3.1.1    Locally weighted linear regression

Daily precipitation amount, Tmean and Trange are estimated for all stations based on the locally weighted linear
regression. Let $x_o$ be the station observation for variable $X$ (precipitation, Tmean, and Trange), the regression estimate
$\hat{x}$ for the target point and time step is obtained as below:

$$x_o = \hat{x} + \varepsilon = \beta_0 + \sum_{i=1}^{n} A_i \beta_i + \varepsilon \qquad (4)$$

where $A_i$ is the $i$th time-invariant attribute (or predictor variables), $\beta_0$ and $\beta_i$ are regression coefficients estimated
using ordinary least squares, and $\varepsilon$ is the residual (or error term). The attributes are latitude, longitude, and elevation
for Tmean and Trange. For precipitation, two more attributes (west-east and south-north slopes) are used to account
for windward and leeward slope precipitation differences. An isotropic Gaussian low-pass filter is used to smooth
DEM before calculating slopes, which can reduce the influence of noise in a high-resolution DEM on the large-scale
topographic effect of precipitation (Newman et al., 2015). Ideally the scale of this smoothing reflects the scale at
which terrain most directly influences precipitation or temperature spatial patterns; in this case the filter bandwidth is
180 km.
For a target station point, $\hat{x}$ is obtained based on data from neighboring stations. Newman et al. (2015, 2019b) used
30 neighboring stations, without controlling for maximum station distance. The very low station density in high-
latitude regions makes this configuration infeasible, hence this study adopts a relatively flexible criterion for selecting
neighboring stations: (1) finding at most 30 stations within a fixed search radius (400 km), and (2) if fewer than 20
stations are found, extending the search radius until 20 stations are found. The least number is set to 20 to ensure that
linear/logistic regression is robust. To incorporate local dependence, a tricube weighting function is used to calculate
the weight $w_{i,j}$ between the target station $i$ and the neighboring station $j$.





$$w_{i,j} = [1 - (\frac{d_{i,j}}{d_{max}})^3]^3 \qquad (5)$$

where $d_{i,j}$ is the distance between $i$ and $j$, and $d_{max}$ depends on the maximum distance ($d_{i,j}^{max}$) between $i$ and all its
neighboring stations. If $d_{i,j}^{max}$ is smaller than 100 km, $d_{max}$ is set to 100 km; otherwise, $d_{max}$ is set to $d_{i,j}^{max}$ + 1 km
(Newman et al., 2015, 2019b). Regression coefficients are estimated by weighted least squares method (described in
in Appendix A).
We found that a small number of observations stations show a climatology that is notably statistically different from
surrounding stations, which could cause an adverse effect on gridded estimates, particularly in sparsely gauged regions.
Strategies are designed to identify and exclude such stations (Appendix B).

### 3.1.2    Locally weighted logistic regression

PoP is estimated using the locally weighted logistic regression by fitting binary precipitation occurrence to spatial
attributes:

$$\text{PoP} = \frac{1}{1 + \exp(-\beta_0 + \sum_{i=1}^{n} A_i \beta_i)} \qquad (6)$$

The attributes ($A_i$) are the same as those used by precipitation regression. Regression coefficients are estimated in
Appendix A.
The errors of precipitation, temperature, and PoP estimates for all stations are calculated as the difference between
regression estimates and station observations using the leave-one-out cross-validation procedure.

### 3.2  Regridding, correction, and merging of reanalysis datasets

The three reanalysis datasets (ERA5, MERRA-2, and JRA-55) have different spatial resolutions and contain
systematic biases. In this section, we discuss steps taken to (1) regrid all reanalysis datasets to the resolution of
EMDNA (0.1°), (2) perform a correction to remove the systematic bias in original estimates, and (3) merge the three
reanalysis datasets to produce a background field that improves over any individual reanalysis dataset, in support of
the reanalysis-station merging described in Sect. 3.3.

### 3.2.1    Regridding of reanalysis datasets

Precipitation, Tmean, and Trange are regridded to 0.1° using locally weighted regression (Fig. 2c). Latitude, longitude,
and elevation are used as predictor variables for simplicity. Precipitation or temperature lapse rates are implicitly
considered by involving elevation in the regression. Raw reanalysis data from a 5 × 5 space window (i.e., 25 coarse-
resolution grids) centered by the 0.1° target grid are used to perform the regression. Each grid is represented using its
center point. This regridding method has been proven effective in previous studies (Xu et al., 2015; Duan and Li, 2016;
Lu et al., 2020). Reanalysis estimates are also regressed to the locations of all stations to facilitate evaluation and



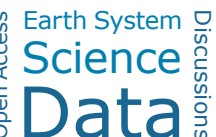

weight estimation in the following steps, which can avoid the scale mismatch caused by using point-scale observations
to evaluate 0.1° gridded estimates (Tang et al., 2018a).
We also tested other regridding methods such as the nearest neighbor, bilinear interpolation, and temperature lapse
rate-based downscaling (Tang et al., 2018b). Results (not shown) indicated that their performance is generally inferior
to the locally weighted regression with respect to several accuracy metrics.

### 209    3.2.2     Probability of precipitation estimation

Reanalysis precipitation can exhibit large biases in the number of wet days because the models often generate many
light precipitation events. To overcome this limitation, we designed two methods for determining the occurrence of
reanalysis precipitation. The first is to use positive thresholds to determine precipitation occurrence. The threshold
was estimated in two ways, namely by forcing reanalysis precipitation (1) to have the same number of wet days with
station data, or (2) to achieve the highest critical success index (CSI). Gridded thresholds can be obtained through
interpolation and used to discriminate between precipitation events or non-events. However, this method can only
obtain binary occurrence instead of continuous PoP between zero and one. The second method is based on univariate
logistic regression. The amount of reanalysis precipitation is used as the predictor and the binary occurrence from
station data is used as the predictand. The logistic regression is implemented for each reanalysis product in the same
way as Sect. 3.1.2. The comparison between the threshold-based method and the logistic regression-based method
shows the latter achieves higher accuracy. Therefore, we adopt the univariate logistic regression to estimate PoP for
each reanalysis product in this study. The possible bias caused by station measurements is not considered.

### 222    3.2.3     Bias correction of reanalysis datasets

Considering reanalysis products usually contain systematic bias (Mooney et al., 2011; Beck et al., 2017; Tang et al.,
2018b, 2020b), the linear scaling method (also known as multiplicative/additive correction factor; Teutschbein and
Seibert, 2012) is used to correct reanalysis precipitation, Tmean, and Trange estimates. Reanalysis PoP is not corrected
because station information has been incorporated in the logistic regression. Let $x_r$ be the reanalysis estimate for
variable $X$, the corrected estimate for a target grid/point $i$ is calculated as:

$$x_{r,i}^* = \begin{cases} x_{r,i} + \dfrac{\sum_{j=1}^m w_{i,j}\left(\bar{x}_{o,j} - \bar{x}_{r,j}\right)}{\sum_{j=1}^m w_{i,j}} & \text{additive correction} \\[3em] x_{r,i} \dfrac{\sum_{j=1}^m w_{i,j} \dfrac{\bar{x}_{o,j}}{\bar{x}_{r,j}}}{\sum_{j=1}^m w_{i,j}} & \text{multiplicative correction} \end{cases} \tag{7}$$

where $x_{r,i}^*$ is the corrected reanalysis estimate, $w_{i,j}$ is the distance-based weight (Eq. (5)), and $\bar{x}_{o,j}$ and $\bar{x}_{r,j}$ are the
climatological mean for each month (e.g., all January from 1979 to 2018) from station observations and reanalysis
estimates for the $j$th neighboring station, respectively. The additive correction is used for Tmean and Trange, and the
multiplicative correction is used for precipitation. The number of neighboring stations ($m$) is set to 10, which is smaller



than that used for linear or logistic regression (Sect. 3.1) but should be enough for bias correction. The upper bound
of $\frac{\bar{x}_{o,j}}{\bar{x}_{r,j}}$ is set to 10 to avoid over-correction in some cases (Hempel et al., 2013).
Linear scaling can also be performed at monthly (Arias-Hidalgo et al., 2013; Herrnegger et al., 2018; Willkofer et al.,
2018) or daily (Vila et al., 2009; Habib et al., 2014) scales by replacing $\bar{x}_{o,j}$ and $\bar{x}_{r,j}$ by monthly mean (e.g., January
in one year) or daily values. We compared the performance of corrections at different scales and found that monthly-
or daily-scale corrections acquire more accurate estimates than the climatological correction. The climatological
correction was adopted because (1) it preserves the absolute/relative trends better than daily or monthly corrections,
and (2) the OI merging (Sect. 3.3) adjusts daily variability of estimates, which compensates for the limitation of
climatological correction and makes daily/monthly-scale correction unnecessary.
Quantile mapping is another widely used correction method (Wood et al., 2004; Cannon et al., 2015). We compared
quantile mapping and linear scaling and found that they are similar in statistical accuracy, while quantile mapping
achieves better probability distributions with much smaller Hellinger distance (Hellinger, 1909) which is a metric used
to quantify the similarity between estimated and observed probability distributions. Nevertheless, quantile mapping
could result in spatial smoothing of precipitation and temperature, particularly in high-latitude regions where stations
are few. For example, Ellesmere Island, the northernmost island of the Canadian Arctic Archipelago, usually shows
lower temperature in inland regions due to orographic uplift. However, quantile mapping will erase this gradient
because reanalysis grids for this island are corrected based on almost the same reference stations. To ensure the
authenticity of spatial distributions, quantile mapping is not used in this study.
**3.2.4    Merging of reanalysis datasets**
The three reanalysis products are merged using the Bayesian Model Averaging (BMA, Hoeting et al., 1999), which
has proved to be effective in fusing multi-source datasets (Chen et al., 2015; Ma et al., 2018a, 2018b). According to
the law of total probability, the PDF of the BMA estimate can be written as:

$$p(E) = \sum_{r=1}^{3} p(E|x_r^*, x_o) \cdot p(x_r^*|x_o) \tag{8}$$

where $E$ is the ensemble estimate, $x_r^*$ ($r$=1, 2, 3) is the bias-corrected estimate from three reanalysis products,
$p(E|x_r^*, x_o)$ is the predicted PDF based only on a specific reanalysis product, and $p(x_r^*|x_o)$ is the posterior probability
of reanalysis products given the station observation $x_o$. The posterior probability $p(x_r^*|x_o)$ can be identified as the
fractional BMA weight $w_r$ with $\sum_{r=1}^{3} w_r = 1$. BMA prediction can be written as the weighted sum of individual
reanalysis products.
For Tmean and Trange, $p(E|x_r^*, x_o)$ can be regarded as the normal distribution $g(E|\theta_r)$ defined by the parameter
$\theta_r = \{\mu_r, \sigma_r^2\}$, where $\mu_r$ is the mean and $\sigma_r^2$ is the variance (Duan and Phillips, 2010). For precipitation, if we apply



Box-Cox transformation (Eq. (3)) to positive events (>0) and exclude zero events, its distribution is approximately
normal, and $p(E|x_r^*, x_o)$ can be represented using $g(E|\theta_r)$. Therefore, Eq. (8) can be written as:

$$p(E) = \sum_{r=1}^{3} w_r \cdot g(E|\theta_r) \tag{9}$$

There are different approaches to infer $w_r$ and $\theta_r$ (Schepen and Wang, 2015). This study uses the log-likelihood
function to estimate the parameters (Duan and Phillips, 2010; Chen et al., 2015; Ma et al., 2018b). The Expectation-
Maximization algorithm (Raftery et al., 2005) can be applied to estimate parameters by maximizing the likelihood
function. BMA weights are obtained for all stations and each month. Gridded weights are obtained using the inverse
distance weighting interpolation.
Merging multiple datasets could affect the probability distributions and extreme characteristics of original datasets.
This is not a major concern because the merged reanalysis data are further adjusted by station data in OI merging (Sect.
3.3), a later step in the EMDNA process. Also, the probabilistic estimation of ensemble members (Sect. 3.4) has a
large effect on estimates of extreme events.
Gridded errors of BMA-merged estimates are necessary to enable optimal interpolation (Sect. 3.3). The error
estimation is realized using a two-layer cross-validation (Appendix C).

### 3.3 Optimal Interpolation-based merging of reanalysis and station data

#### 3.3.1    Optimal Interpolation

OI has proven to be effective in merging multiple datasets (Sinclair and Pegram, 2005; Xie and Xiong, 2011) and has
been applied in operational products such as CaPA (Mahfouf et al., 2007; Fortin et al., 2015) and the China Merged
Precipitation Analysis (CMPA, Shen et al., 2014, 2018). Let $x_A$ be the OI analysis estimate. The OI analysis estimate
($x_{A,i}$) for a target grid/point $i$ and time step is obtained by adding an increment to the first guess of the background
($x_{B,i}$). The increment is a weighted sum of the difference between observation and background values at neighboring
stations.

$$x_{A,i} = x_{B,i} + \sum_{j=1}^{m} w_j(x_{O,j} - x_{B,j}) \tag{10}$$

where $x_{O,j}$, $x_{B,j}$, and $w_j$ are the observed value (subscript $O$), background value (subscript $B$), and weight for the $j$th
neighboring station. Let $x_T$ be the true value, the errors of observed and background values are $\varepsilon_{O,j} = x_{O,j} - x_{T,j}$ and
$\varepsilon_{B,j} = x_{B,j} - x_{T,j}$ (or $\varepsilon_{B,i} = x_{B,i} - x_{T,i}$), respectively. Assuming that (1) the observation and background errors are
unbiased with an expectation of zero and (2) there is no correlation between background and observation errors, the
weights that minimize the variance of the analysis errors can be obtained by solving:





$$w(R + B) = b \qquad (11)$$

where $\mathbf{w}$ is the vector of $w_j$ ($j = 1, 2, ..., m$), $\mathbf{R}$ and $\mathbf{B}$ are $m \times m$ covariance matrices of $\varepsilon_{O,j}$ and $\varepsilon_{B,j}$, respectively, and $\mathbf{b}$ is the $m \times 1$ vector of covariance between $\varepsilon_{B,i}$ and $\varepsilon_{B,j}$. The background provided by reanalysis models assimilates observations in the production and is corrected in a way using station data (described in Sect. 3.2.3), which may affect the soundness of the second assumption. The effect of this slight violation, however, is rather small according to our results and previous studies (Xie and Xiong, 2011; Shen et al., 2014b, 2018).

Different approaches can be used to implement OI. For example, Fortin et al. (2015) use raw station observations as $x_O$, and assumes that the background error is a function of error variance and correlation length, and the observation error is a function of error variance. The variances and correlation length are obtained by fitting a theoretical variogram using station observations. Xie and Xiong (2011) and Shen et al. (2014) use station-based gridded estimates as $x_O$, and assume that the background error variance is a function of precipitation intensity, the cross-correlation of background errors is a function of distance, and the observation error variance is a function of precipitation intensity and gauge density. The parameters of those functions are estimated based on station data in densely gauged regions.

In this study, we adopt a novel design that calculates weights based on error estimation, a feature that is enabled by the probabilistic nature of the observational dataset. Regression estimates and their errors at station points (Sect. 3.1) are used as $x_O$ and $\varepsilon_O$, respectively. BMA-merged reanalysis estimates and their errors (Sect. 3.2) are used as $x_B$ and $\varepsilon_B$, respectively. We do not use gridded regression estimates because (1) $x_{O,j} - x_{B,j}$ will show weak variation if neighboring stations are replaced by neighboring grids, and (2) estimates of weights $\mathbf{w}$ could be unrealistic because of the spatial smoothing of interpolated regression errors. The advantages of this design are (1) weights and inputs closely match each other and (2) weights in sparsely gauged regions are not determined by parameters fitted in densely gauged regions.

The Box-Cox transformation is applied to precipitation estimates. Then, precipitation, PoP, Tmean, and Trange estimates provided by OI are used as $\mu$ and PoP required for generating meteorological ensembles.

### 3.3.2    Error of OI-merged estimates

Variance is a necessary parameter to enable ensemble estimation. The variance $\sigma^2$ is represented using the mean squared error of OI estimates in this study. First, the error of OI analysis estimates ($\varepsilon_A = x_A - x_o$) is obtained for all stations using the leave-one-out strategy. Then, the $\sigma_i^2$ for the $i$th grid is obtained as a weighted sum of squared errors from neighboring stations:

$$\sigma_i^2 = \frac{\sum_{j=1}^m w_{i,j} (\varepsilon_{A,j})^2}{\sum_{j=1}^m w_{i,j}} \qquad (12)$$

where $\varepsilon_{A,j}$ is the difference between the station observation and OI estimate at the $j$th neighboring station, and $w_{i,j}$ is the weight (Eq. (5)).

### 3.3.3 Correction of precipitation under-catch

Considering station precipitation data usually contain measurement errors such as wind-induced under-catch particularly in high-latitude and mountainous regions, OI-merged precipitation is further adjusted using the bias-corrected precipitation climatology produced by Beck et al. (2020). This climatology infers the long-term precipitation using a Budyko curve and streamflow observations. Three corrected datasets are provided, including WorldClim, version 2 (WorldClim V2; Fick and Hijmans, 2017), the Climate Hazards Group Precipitation Climatology, version 1 (CHPclim V1; Funk et al., 2015) and Climatologies at High Resolution for the Earth's Land Surface Areas, version 1.2 (CHELSA V1.2; Karger et al., 2017). The water balance-based method of Beck et al. (2020) considers all measurement errors (e.g., under-catch and wetting/evaporation loss) as a whole and under-catch is the major error source in many regions.

Although the three datasets show similar precipitation distributions after bias correction, CHELSA V1.2 is used because its period (1979–2013) is most similar to our study period (1979–2018). The correction of OI-merged precipitation is performed in two steps: (1) the ratio between bias-corrected CHELSA V1.2 and OI-merged long-term monthly precipitation is calculated at the 0.1° resolution during 1979–2013, and (2) daily OI-merged precipitation estimates during 1979–2018 are scaled using the corresponding monthly ratio map. The bias correction notably increases precipitation in northern Canada and Alaska (Fig. 2d) where under-catch of precipitation is often large.

### 3.4 Ensemble generation

#### 3.4.1 Spatiotemporally correlated random fields

Spatially correlated random fields (SCRFs) are used to sample from the probability distributions of precipitation and temperature. The SCRFs are produced using the following three steps. First, the spatial correlation structure is generated based on an exponential correlation function:

$$c_{i,j} = \exp\left(-\frac{d_{i,j}}{C_{len}}\right) \tag{13}$$

where $d_{i,j}$ is the distance between grids $i$ and $j$, and $C_{len}$ is the spatial correlation length determined for each climatological month based on regression using station data for precipitation, Tmean, and Trange, separately. The spatial correlation structure is generated using the conditional distribution approach. Every point is conditioned on previously generated points which are determined using a nested simulation strategy to improve the calculation efficiency (Clark and Slater, 2006).

Second, the spatially correlated random field ($\mathbf{R}_t$) for the $t$th time step is generated by sampling from the normal distribution with the mean value and standard deviation depending on the random numbers of previously generated grids (Clark and Slater, 2006).



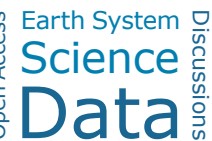

Third, the SCRF is generated by incorporating spatial and temporal correlation relationships. Let $\rho_{TM}$ and $\rho_{TR}$ be the
lag-1 auto-correlation for Tmean and Trange, respectively, $\rho_{CR}$ be the cross-correlation between Trange and
precipitation, $\mathbf{R}_{t-1,TM}$, $\mathbf{R}_{t-1,TR}$ and $\mathbf{R}_{t-1,PR}$ be the SCRF for the (*t-1*)th time step for Tmean, Trange, and precipitation,
respectively, the SCRF for *t*th time step following (Newman et al., 2015) is written as:

$$
\begin{cases}
\mathbf{R}_{t,TM} = \rho_{TM}\mathbf{R}_{t-1,TM} + \sqrt{1 - \rho_{TM}^2}\mathbf{R}_{t-1,TM} \\[2mm]
\mathbf{R}_{t,TR} = \rho_{TR}\mathbf{R}_{t-1,TR} + \sqrt{1 - \rho_{TR}^2}\mathbf{R}_{t-1,TR} \\[2mm]
\mathbf{R}_{t,PR} = \rho_{CR}\mathbf{R}_{t-1,TR} + \sqrt{1 - \rho_{CR}^2}\mathbf{R}_{t-1,PR}
\end{cases}
\tag{14}
$$

### 349    3.4.2    Probabilistic estimation

Probabilistic estimates are produced using the probability distribution $N(\mu, \sigma^2)$ in Eq. (1) and $\mathbf{R}$ in Eq. (14). For
Tmean and Trange, the SCRF ($\mathbf{R}_{TM}$ and $\mathbf{R}_{TR}$) is directly used as the standard normal deviate ($R_X$). The estimate ($x_e$)
for the ensemble member *e* is written as:

$$
x_e = \mu + R_X \cdot \sigma
\tag{15}
$$

For precipitation, an additional step is to judge whether an event occurs or not according to OI-merged PoP and the
estimated probability from the SCRF. Let $F_N(x)$ be the CDF of the standard normal distribution, $F_N(R_{PR})$ is the
cumulative probability corresponding to the random number $R_{PR}$. If $F_N(R_{PR})$ is larger than $p_0$, the scaled cumulative
probability of precipitation ($p_{cs}$) is calculated as:

$$
p_{cs} = \frac{F_N(R_{PR}) - p_0}{1 - p_0}
\tag{16}
$$

The probabilistic estimate for precipitation can be expressed as:

$$
x_e = \begin{cases}
0 & if \quad F_N(R_{PR}) \leq p_0 \\
\mu + F_N^{-1}(p_{cs}) \cdot \sigma & if \quad F_N(R_{PR}) > p_0
\end{cases}
\tag{17}
$$

### 358    3.5 Evaluation of probabilistic estimates

Independent stations that are not used in SCDNA are used to evaluate EMDNA because the leave-one-out strategy is
too time-consuming for evaluating probabilistic estimates. GHCN-D stations with precipitation or temperature records
less than eight years are extracted because SCDNA restricts attention to stations with at least eight-year records. In
total, 15,018 precipitation stations and 2,455 temperature stations are available for independent testing.





The Brier skill score (BSS; Brier, 1950) is used to evaluate probabilistic precipitation estimates. The continuous ranked
probability skill score (CRPSS) is used to evaluate probabilistic temperature estimates. Their definitions are described
in Appendix D.
Furthermore, the reliability and discrimination diagrams are used to assess the behavior of probabilistic precipitation
estimates. The reliability diagram shows the conditional probability of an observed event (precipitation above a
threshold) given the probability of probabilistic precipitation estimates. In a reliability diagram, a perfect match has
all points located on the 1-1 line. The discrimination diagram shows the PDF of probabilistic precipitation estimates
for different observed categories. For precipitation, two categories are defined: events or non-events, i.e., observed
precipitation above or below a threshold. The difference between PDF curves of events or non-events represents the
degree of discrimination. Larger discrimination is preferred. The PDF for non-event/event should be maximized at the
probability of zero/one.
**4.  Results**
**4.1 Comparison between raw and merged reanalysis estimates**
The three raw reanalysis estimates are regridded, corrected for bias, and merged. In this section, we directly compare
raw and BMA-merged estimates. The evaluation is performed for all stations using the two-layer cross-validation
strategy. The correlation coefficient (CC) and root mean square error (RMSE) are used as evaluation metrics.
For precipitation, the three reanalysis products show the highest CC in CONUS and the lowest CC in Mexico (Fig. 3).
The slight spatial discontinuity of CC along the Canada-USA border and the USA-Mexico border (Fig. 3 and 6) is
caused by the inconsistent reporting time of stations. Daily precipitation from reanalysis products is accumulated from
0 to 24 UTC, while stations from different countries or regions usually have different UTC accumulation periods
(Beck et al., 2019; Tang et al., 2020a). The distributions of RMSE agrees with those of precipitation amounts with
higher values in the southern corner and west coast of North America and western CONUS. Overall, ERA5
outperforms MERRA-2 followed by JRA-55.
BMA-merged precipitation estimates show higher accuracy than all reanalysis products (Fig. 3). For ERA5 and JRA-
55, the improvement of CC and RMSE is the most evident in the Rocky Mountains, while for MERRA-2, the largest
improvement occurs in central CONUS. ERA5 is the closest to BMA estimates concerning CC and RMSE. The
improvement of BMA estimates against ERA5 is more prominent in the high-latitude regions. Specifically, the mean
CC increases by 0.05 and 0.07 in regions southern and northern to 55°N, respectively. The corresponding decrease of
mean RMSE is 0.72 and 0.89 mm/d, respectively.

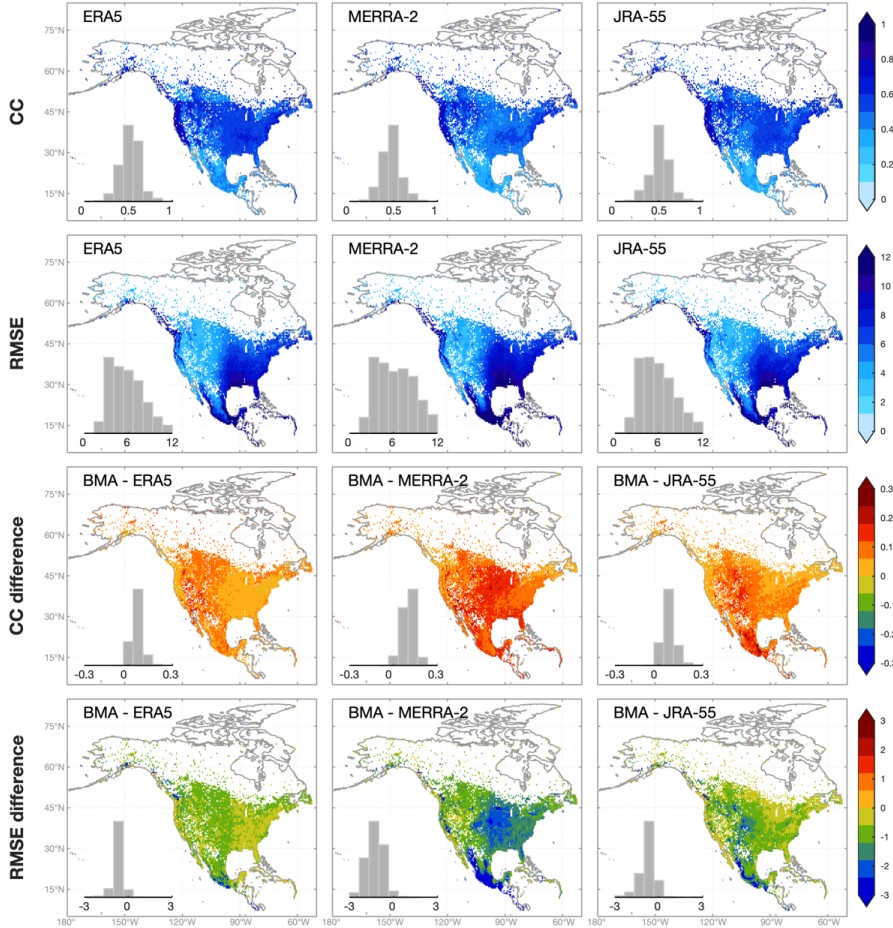

Figure 3. The spatial distributions and histograms of CC (the first row) and RMSE (the second row) based on raw
reanalysis precipitation estimates (ERA5, MERRA-2, and JRA-55). The improvement of BMA-merged estimates
against raw reanalysis estimates is shown in the third and fourth rows. The grid resolution is 0.5°. For each 0.5° grid
point, the median value of all stations located within the grid is shown.

The CC of reanalysis Tmean estimates is close to one in most regions of North America (Fig. 4) and still above 0.9 in
Mexico where the CC is the lowest. According to RMSE, Tmean estimates have the largest error in western North
America because coarse-resolution raw reanalysis estimates cannot reproduce the variability of temperature caused
by elevation variations. The rank of three reanalysis products for Tmean is the same as that for precipitation with
ERA5 being the best one. BMA estimates show higher CC than reanalysis products particularly in Mexico, while the
improvement of RMSE is the most notable in the Rocky Mountains. For a few stations, the RMSE of BMA estimates

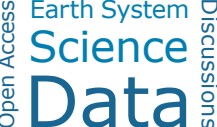

is slightly worse than raw reanalysis estimates (Fig. 4) because the downscaling of reanalysis temperature could
occasionally magnify the error in low-altitude regions (Tang et al., 2018b).
For Trange, BMA estimates show much larger improvement than Tmean, while the differences of CC and RMSE are
relatively evenly distributed (Fig. 5). The improvement of BMA estimates against JRA-55 estimates is especially large.
In general, BMA is effective in improving the accuracy of reanalysis precipitation and temperature estimates.

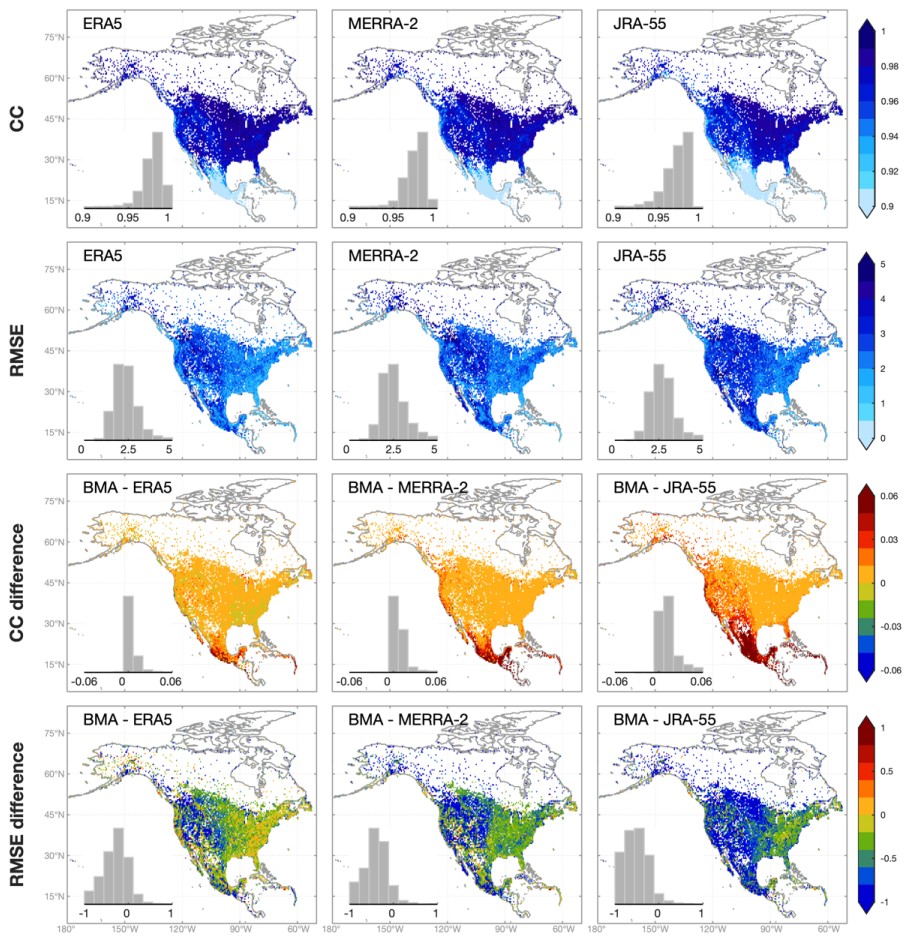


Figure 4. Same with Figure 3, but for mean temperature.

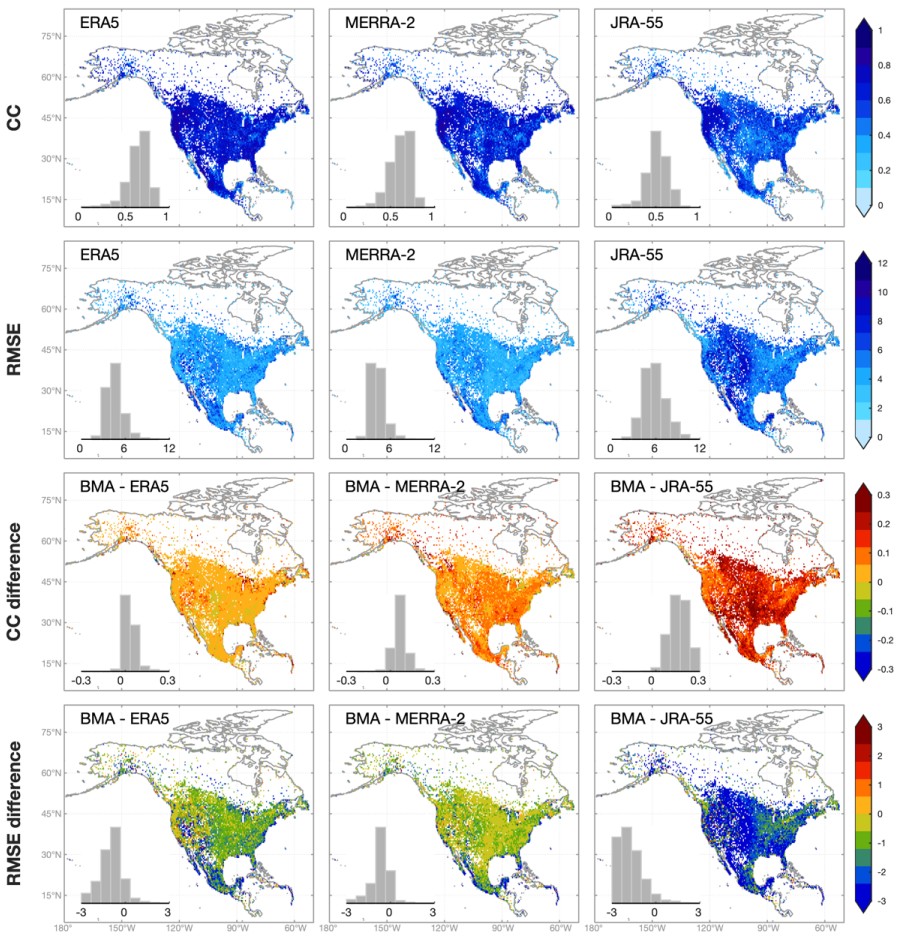


Figure 5. Same with Figure 3, but for daily temperature range.


**4.2 The performance of optimal interpolation**


Optimal interpolation is used to combine station-based estimates with reanalysis estimates. The performance of OI-
merged precipitation and temperature estimates is compared to the background (BMA-merged reanalysis estimates;
Fig. 6) and observation (station-based regression estimates; Fig. 7) inputs. To better show the spatial variations of the
improvement of OI estimates, RMSE for precipitation and Trange is normalized using the mean value (termed as
NRMSE), while Tmean is evaluated using RMSE.
Overall, OI estimates are more accurate than merged reanalysis or station regression estimates for all variables across
North America. Comparing OI estimates to reanalysis estimates, for precipitation, Tmean, and Trange, the mean CC
is improved by 0.24, 0.02, and 0.15, respectively, and the mean RMSE is reduced by 1.88 mm/d, 0.52℃, and 0.87℃,



respectively. The improvement of OI estimates against station estimates is smaller with the mean CC increasing by
0.06, 0.01 and 0.05, and the mean RMSE decreasing by 0.56 mm/d, 0.18℃, and 0.29℃ for precipitation, Tmean, and
Trange, respectively.
OI can utilize the complementarity between station and reanalysis estimates. For example, according to CC, the
improvement of OI estimates against reanalysis estimates is larger in the eastern than the western CONUS, while the
improvement against station estimates is larger in western than eastern CONUS. This means that although station
estimates generally show higher accuracy reanalysis estimates, station estimates face more severe quality degradation
in mountainous regions. Moreover, the latitudinal curves of CC and NRMSE in Fig. 6 and 7 indicate that the
improvement of OI estimates against reanalysis estimates decreases as the latitude increases from southern CONUS
to northern Canada, while the improvement against station estimates shows a reverse trend.
For Tmean, the CC improvement for OI estimates is the largest in Mexico and decreases from low to high latitudes,
while based on RMSE, the improvement increases with latitude. For Trange, the latitudinal variation exhibits a similar
pattern with precipitation for regions north of 50°N, with larger/smaller improvement in higher latitudes against
station/reanalysis estimates. For regions south of 50°N, the improvement of CC and NRMSE against station estimates
shows different trends.
The latitudinal variations in Fig. 6 and 7 are related to station densities (Fig. 8). Station-based estimates often have
lower accuracy in regions with scarce stations (i.e., high-latitude North America), while reanalysis estimates could
have less dependence on station densities due to the compensation of physically-based models. For precipitation, the
improvement of OI estimates against regression estimates increases with the distance according to both CC and
NRMSE, while the improvement against reanalysis estimates shows an inverse trend (Fig. 8). The shaded area figure
within Fig. 8 shows that most stations can find the 20 neighboring stations within the search radius of 20-100 km.
However, as the distance increases beyond 200 km, the number of stations becomes very small while the number of
grids is still large. For Tmean, the trend with distance is not obvious probably because it is usually easier to interpolate
Tmean observations due to its strong linkage with elevation and latitude. For Trange, the improvement against
reanalysis and station estimates both increases with the distance. The results show that OI merging is particularly
useful in sparsely gauged regions.

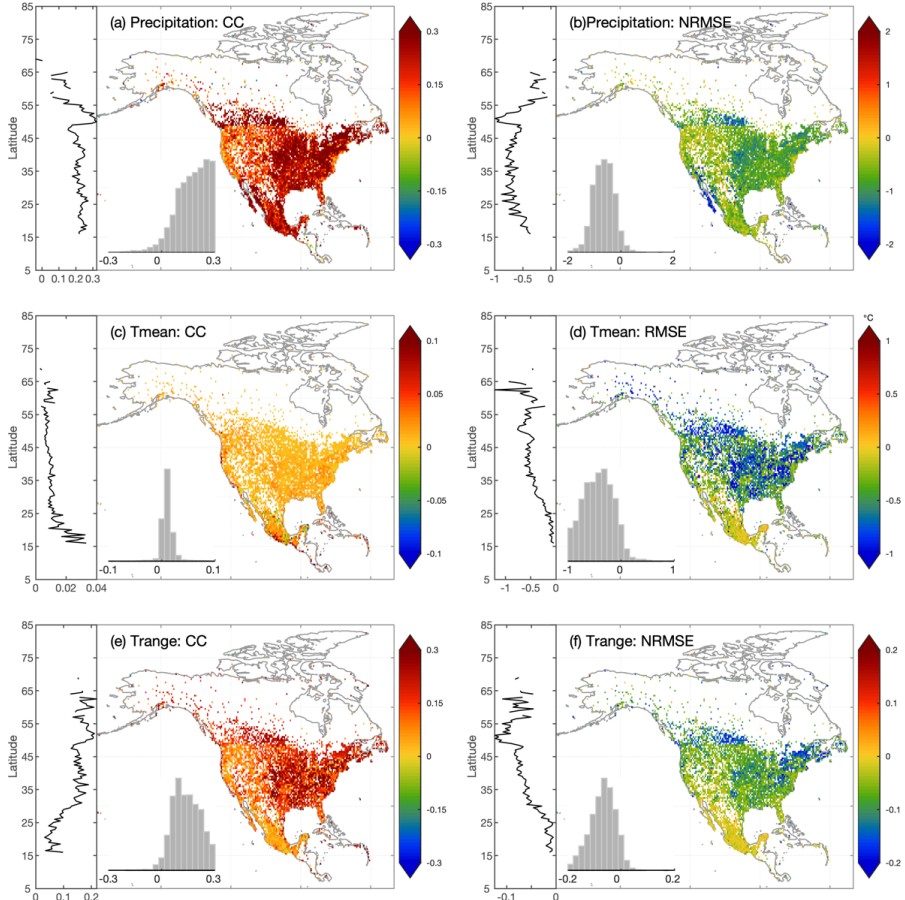

Figure 6. The differences of (a) CC and (b) NRMSE (normalized RMSE) between OI-merged precipitation estimates
and BMA-merged reanalysis precipitation estimates. The latitudinal distributions of metrics are attached on the left
side, showing the median value for 0.5° latitude bands. (c-d) are the same with (a-b) but for mean temperature and
RMSE is not normalized. (e-f) are the same with (a-b) but for daily temperature range.

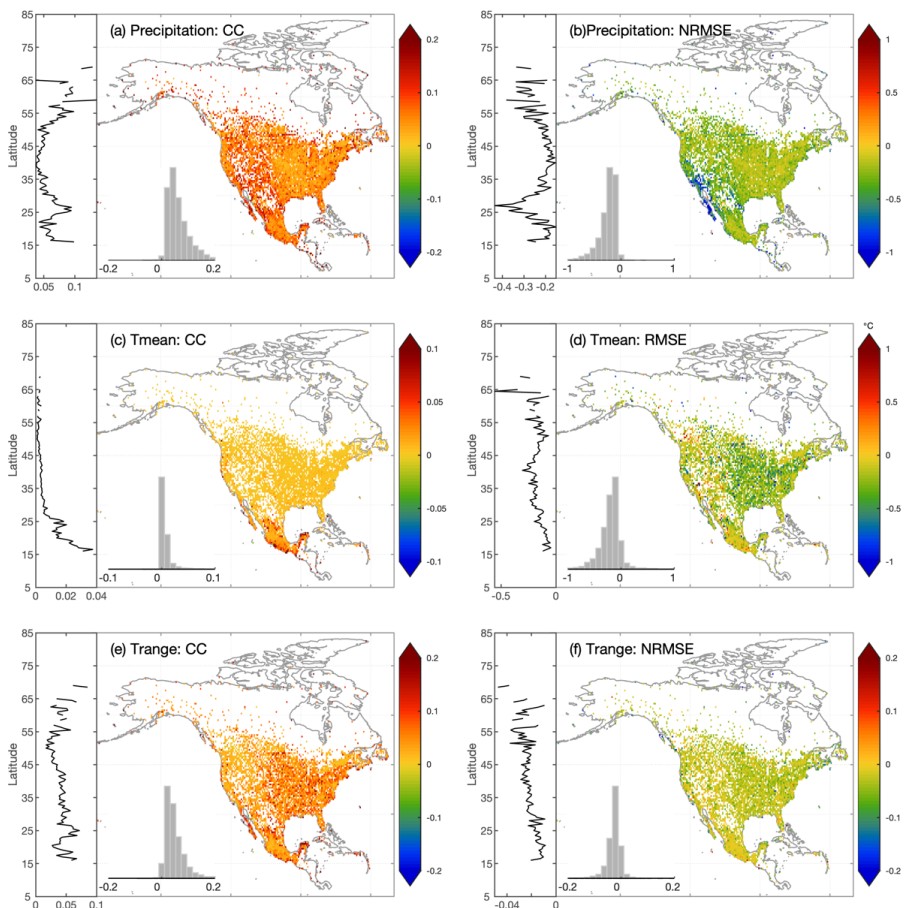


Figure 7. Similar with Figure 6, but the differences are between OI-merged precipitation estimates and station-based
regression precipitation estimates.

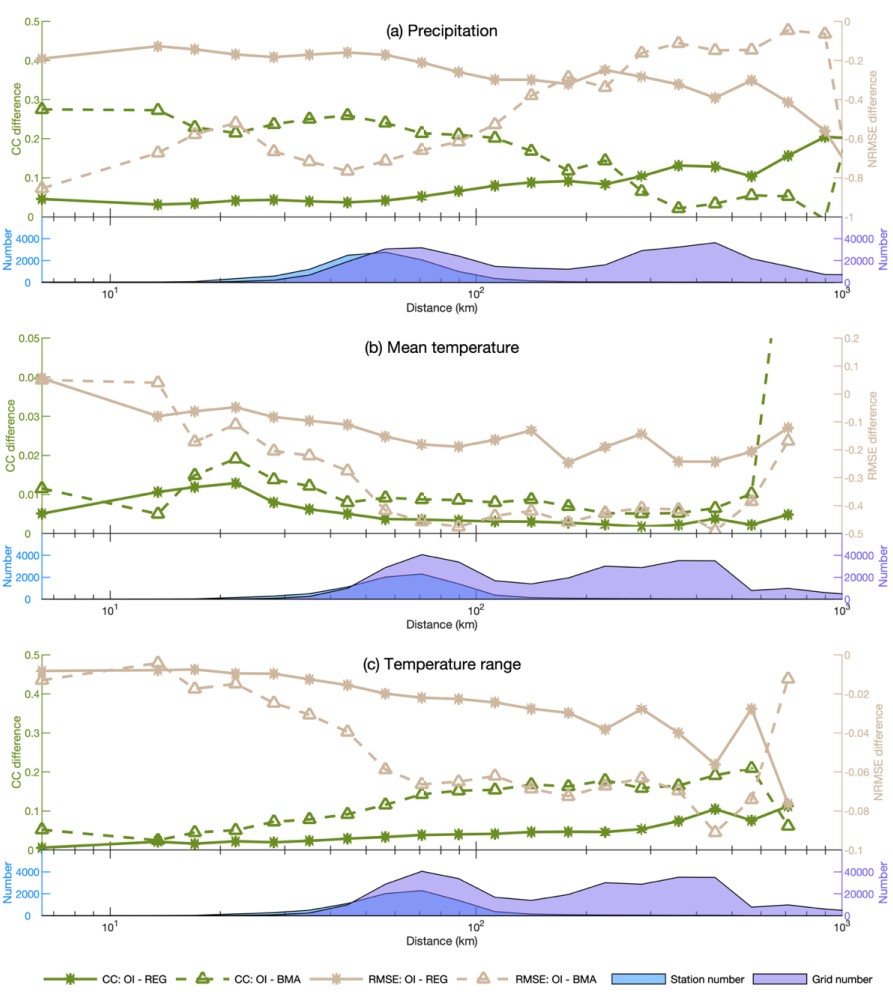


Figure 8. The improvement of OI-based station-reanalysis merged estimates against station-based regression (REG) and BMA-merged reanalysis (BMA). The logarithmic X-axis shows the distance between the target station/grid and its 20th distant neighboring station. A larger distance represents a lower station density. The shaded area chart shows the numbers of stations and grid points corresponding to the same distance, which is the same for mean temperature and temperature range.

## 4.3 Evaluation of probabilistic estimates

The distributions of the OI and ensemble precipitation, Tmean, and Trange estimates in June 2016 are shown in Fig. 9. Compared with OI precipitation estimates, ensemble precipitation estimates show generally consistent but less smooth distributions because of the relatively short spatial correlation length in the warm season. For Tmean and

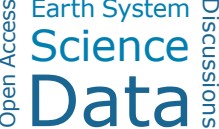

Trange, OI and ensemble estimates show very similar spatial distributions. Precipitation shows the largest standard
deviation, while Tmean shows the smallest, because the standard deviation is determined by the errors of OI estimates.
The PoP from station observations and ensemble estimates is compared based on stations with at least 5-year-long
records from 1979 to 2018 (Fig. 10). The comparison cannot represent climatological PoP (Newman et al., 2019b)
due to short time length of independent stations (Sect. 3.5). Overall, EMDNA estimates show similar PoP distributions
with station observations. The PoP in Canada is slightly overestimated because (1) the quality of EMDNA is lower in
regions with fewer stations and (2) point-scale station observations could underestimate the PoP at a larger scale (e.g.,
0.1° grids) as shown by Tang et al. (2018a).

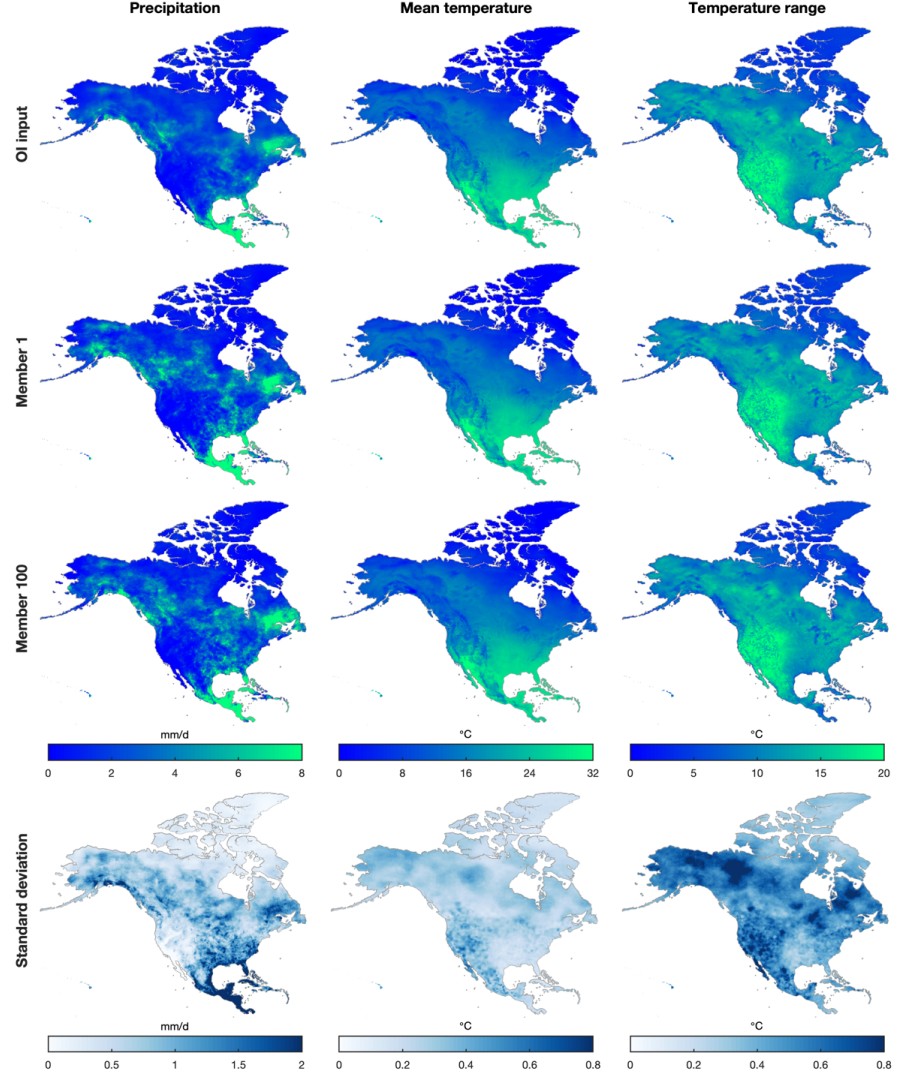




Figure 9. The distributions of average values from precipitation (the first column), mean daily temperature (the second column), and daily temperature range (the third column) averaged over the period 1-30 June 2016. The first to third rows represent estimates from OI-merged inputs, ensemble member 1, and ensemble member 100. The fourth row represents the standard deviation of all the 100 members for one month (June 2016).

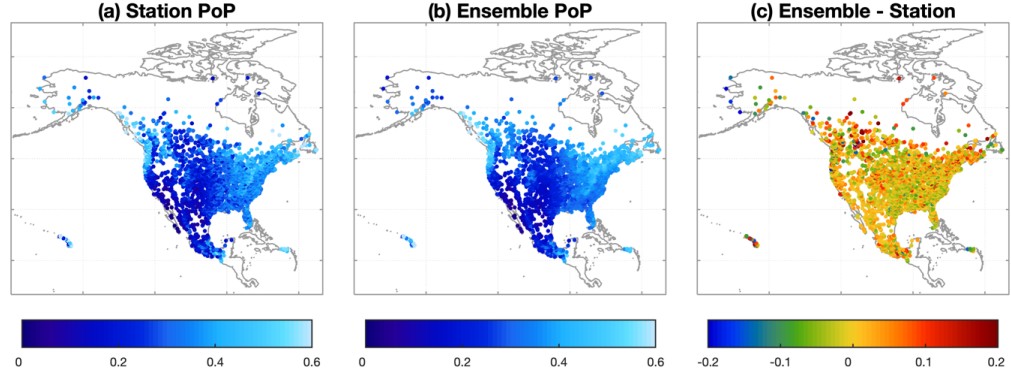

Figure 10. The probability of precipitation (PoP) from (a) station observations and (b) concurrent EMDNA ensemble estimates with their differences shown in (c). Stations with at least 5-year-long records from 1979 to 2018 are involved in the comparison.

The discrimination diagram (Fig. 11) shows that ensemble precipitation assigns the highest occurrence frequency at the lowest estimated probability for non-precipitation events, and the performance becomes better as the threshold increases from 0 to 50 mm. For precipitation events, ensemble estimates show the highest frequency at the highest estimated probability for the thresholds of 0, 10, and 25 mm, while as the threshold increases, the frequency curve becomes skewed to the lower estimated probability. This problem is also seen in Clark and Slater (2006) and Newman et al. (2015). Ensemble precipitation shows good reliability for all precipitation thresholds with the points located at or close to the 1-1 line (Fig. 11). At low and high estimated probabilities of occurrence, ensemble precipitation shows slight wet bias. The reliability performance does not show clear dependence with thresholds.

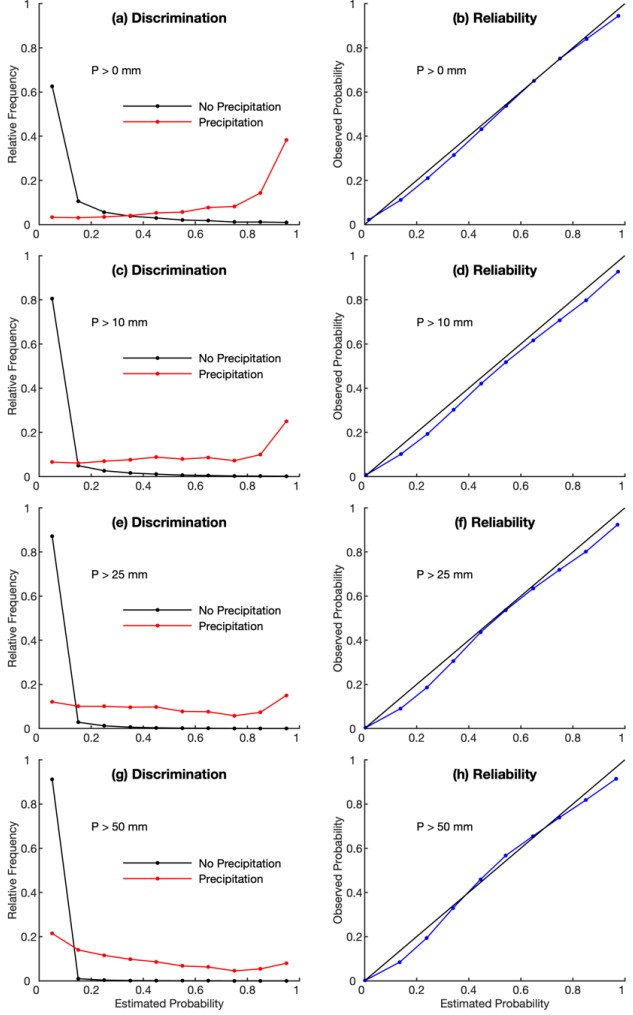


Figure 11. The discrimination and reliability diagrams based on ensemble precipitation estimates. Four rain/no rain thresholds (0, 10, 25, 50 mm) are used.

The BSS for precipitation and CRPSS for Tmean and Trange are shown in Fig. 12. In most cases, ensemble
precipitation shows the highest frequency when BSS is above 0.5. As the precipitation threshold increases, the BSS
values decrease. The median BSS values are 0.62, 0.54, and 0.46 for the thresholds of 0, 10, and 20 mm/d, respectively.
We note that a small number of cases show BSS values smaller than zero, indicating that the ensemble estimated
probability is worse than climatological probability. A low BSS value usually occurs in regions where precipitation is
hard to estimate (e.g., Rocky Mountains) resulting in inaccurate parameters of Eq. (1).



The BSS for all thresholds shows a clear increasing trend from 1979 to 2018 (Fig. 12b) because the observed
precipitation samples from SCDNA increase during this period (Fig. 2 in Tang et al. (2020b)). The increasing trend
of BSS is particularly prominent from 2003 to 2009, during which precipitation samples in the USA experience the
greatest increase (Tang et al., 2020a). The results show that although infilled station data contribute to higher station
densities, observation samples still have a significant effect on gridded data estimation.
Tmean shows high CRPSS for most cases with the frequency peak occurring at ~0.8. The CRPSS of Trange is much
lower with the peak occurring at ~0.6. The median CRPSS for Tmean and Trange is 0.74 and 0.51, respectively.
Analyses show that among stations with negative CRPSS, most are located in Mexico due to the degraded quality of
temperature estimates (Sect. 4.1 and 4.2). The long-term variation of CRPSS is not shown because independent
temperature stations are insufficient to support validation between 1986 and 2010.

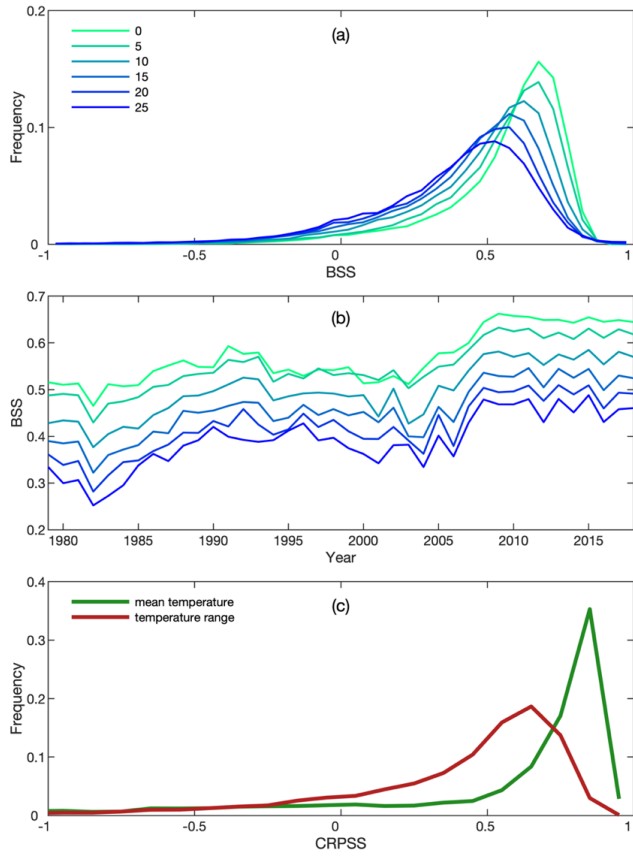


Figure 12. (a) The frequency distributions of the Brier Skill Score (BSS) for precipitation corresponding to rain/no
rain thresholds from 0 to 25 mm/d. (b) The distributions of BSS for precipitation from 1979 to 2018. For each year,





the median value of all stations is used. (c) The frequency distributions of the continuous ranked probability skill score (CRPSS) for daily mean temperature and daily temperature range.

## 5. Discussion

This study presents the framework for producing an ensemble precipitation and temperature dataset over North America. Although we have tested multiple choices of methods (Sect. 3) and overall the product shows good performance (Sect. 4), the methodology still has limitations that need to be improved through continued efforts.

### 5.1 Implementation of OI

OI is used to merge reanalysis outputs and station data. To implement OI-based merging, a critical step is to estimate the weights. Previous studies usually adopt empirical error or variogram functions and fit the parameters using station observations (e.g., CaPA (Fortin et al., 2015) and CMPA (Shen et al., 2018)); then the parameters are constant for the whole study area in the actual application.

In this study, we proposed a novel design, which uses station-based regression estimates as the observation filed and calculates weights by directly solving the weight functions based on observation and background errors. Compared with methods that use station data as the observation field, our method is characterized by inferior estimation of the observation field but realistic estimation of weights. The close linkage between the observation field and the weights could benefit OI estimates but comparing different OI implementations is still meaningful and necessary considering OI has been widely used and is the core algorithm of some operational products.

Furthermore, regression estimates show worse performance in regions with fewer stations. More advanced interpolation methods that can utilize climatology information and comprehensively consider topographic and atmospheric conditions (Daly et al., 2008; Newman et al., 2019b; Newman and Clark, 2020) should be examined in future studies.

### 5.2 Probabilistic estimation

Power transformations (e.g., Box-Cox and root/cubic square) with fixed parameters have proven to be useful in precipitation estimation and dataset production (Fortin et al., 2015, 2018; Cornes et al., 2018; Khedhaouiria et al., 2020; Newman et al., 2020). The Box-Cox transformation with a constant parameter is applied following Fortin et al. (2015) and Newman et al. (2019b, 2020). A fixed parameter, however, cannot ensure that transformed precipitation is normally distributed everywhere as is desirable.

We tested a series of additional parametric and non-parametric transformations based on power functions, logarithmic functions, or a mix of both, and optimized the parametric transformation functions (including Box-Cox) for every grid by minimizing the objective function which is the sum of squared L-skewness and L-kurtosis (Papalexiou and Koutsoyiannis, 2013). Theoretically, compared to a Box-Cox transformation with a fixed parameter, the optimized



functions can obtain precipitation series closer to the normal distribution which should benefit probabilistic estimation,
while the evaluation results show that the Box-Cox transformation with a fixed parameter is better at probabilistic
estimation than optimized functions. We suggest there are three reasons for this: (1) the standard deviation in Eq. (1)
is obtained by interpolating OI errors (Sect. 3.2.2) from neighboring stations, whereas the optimized transformation
parameters could be different at those stations, (2) zero precipitation is excluded during optimization to avoid invalid
transformation or optimization, which reduces the number of stations for every time step and thus degrades the quality
of the spatial interpolation, and (3) the errors caused by back transformation could be large if the optimized
transformation is too powerful. More efforts are needed to resolve this problem.
There are other potential directions for improvement. For example, SCRF is generated from Gaussian distributions,
while other choices such as copulas functions (Papalexiou and Serinaldi, 2020) show potential in probabilistic
estimation. The spatial correlation length is constant for the whole study area following Newman et al. (2015, 2019b),
which may introduce uncertainties for a large domain. Overall, studies related to the production of ensemble
meteorological datasets are still insufficient, particularly for large areas. More studies are needed to clarify the critical
issues in large-scale probabilistic estimation and explore the effect of parameter/method choices on probabilistic
estimates.

### 558    5.3  Alternate data sources

The quality of source data (station observations and reanalysis models) primarily determines the quality of output
datasets. The density of stations has a critical effect on the accuracy of the observation field and probabilistic estimates.
While SCDNA collects data from multiple datasets, efforts are ongoing to expand the database by involving station
sources such as provincial station networks in Canada.
For reanalysis products, ERA5, MERRA-2, and JRA-55 are regridded using locally weighted linear regression to meet
the target resolution. There are some choices for future improvement, such as (1) adopting/developing better
downscaling methods or (2) utilizing outputs from high-resolution re-analysis products or forecasting models such as
ERA5-Land or the Weather Research and Forecasting (WRF) model. For the latter one, a comprehensive assessment
of available products is necessary before substituting the three reanalysis products used by EMDNA. Moreover,
including other data sources such as satellite (e.g. GPM-IMERG) and weather radar estimates is also an opportunity.

### 569    5.4  Precipitation under-catch

Although station precipitation observations are used as the reference in this study, these values are subject to
measurement errors such as wetting loss, wind-induced under-catch, and trace precipitation. Under-catch of
precipitation is particularly severe in high latitudes and mountains due to the stronger wind and frequent snowfall
(Sevruk, 1984; Goodison et al., 1998; Nešpor and Sevruk, 1999; Yang et al., 2005; Scaff et al., 2015; Kochendorfer
et al., 2018). For example, underestimation of precipitation could be larger than 100% in Alaska (Yang et al., 1998).
Bias correction of station precipitation data should consider many factors such as gauge types, precipitation phase,



and environmental conditions, which would be very complicated when a large number of sparsely distributed stations
are involved over the whole of North America.
The under-catch correction used in this study relies on bias-corrected precipitation climatology produced by Beck et
al. (2020), which infers the long-term precipitation using a Budyko curve and streamflow observations. The bias-
corrected precipitation climatology, however, is less accurate in northern Canada where streamflow stations are few
(Beck et al., 2020). In addition, the streamflow data used by the bias-corrected climatology also contain uncertainties
(Hamilton and Moore, 2012; Kiang et al., 2018) related to factors such as streamflow derivation methods (e.g., rate
curves) and measurement instruments. Whilst various under-catch correction methods (e.g., Fuchs et al., 2001; Beck
et al., 2020; Newman et al., 2020) exist, further studies are needed to compare these solutions considering their
effectiveness and availability of input data in a large domain.

## 6. Data availability

The EMDNA dataset is available at https://doi.org/10.20383/101.0275 (Tang et al., 2020a) in netCDF format.
Individual ensemble member, ensemble mean, and ensemble spread of precipitation, Tmean, and Trange are provided.
The total data size is 3.35 TB. Since the 100 members are equally plausible, users can download fewer members if the
storage space and processing time are limited.
The deterministic OI estimates of precipitation, PoP, Tmean, and Trange produced in this study are also available in
netCDF format. The high-quality OI estimates merge reanalysis and station data, which can be useful to applications
that do not need ensemble forcings. The total data size is 40.84 GB.
A teaser dataset of probabilistic estimates is provided to facilitate easy preview of EMDNA without downloading the
entire dataset. The teaser dataset covers the region from -116.8° to -115.2°W, and 50.7° to 51.9°N, the time from 2014
to 2015, and the ensemble members from 1 to 25. The total data size is smaller than 30 MB. See Appendix E for a
brief introduction.

## 7. Summary and Conclusions

Ensemble meteorological datasets are of great value to hydrological and meteorological studies. Given the lack of a
historical ensemble dataset for the entire North America, this study develops EMDNA by integrating multi-source
information to overcome the limitation of sparse stations in high-latitude regions. EMDNA contains precipitation,
Tmean, and Trange estimates at 0.1° spatial resolution and daily temporal resolution from 1979 to 2018 with 100
members. Multiple methodological choices are examined when determining critical steps in the production of
EMDNA. The ultimate framework composes of four main steps: (1) generating station-based interpolation estimates
from SCDNA using locally weighted linear/logistic regression, (2) regridding, correction, and merging of reanalysis
products (ERA5, MERRA-2, and JRA-55), (3) merging station-reanalysis estimates using OI based on a novel method



of OI weight calculation, and (4) generating ensemble estimates by sampling from the estimated probability
distributions with the perturbations provided by SCRF.
The performance of each step is comprehensively evaluated using multiple methods. The results show that the design
of the framework is effective. In short, we find that (1) station-based interpolation estimates are less accurate in regions
with sparse stations (e.g., high latitudes) and complex terrain; (2) BMA-merged reanalysis estimates show notable
improvement against raw reanalysis estimates, particularly for precipitation and Trange and over high-latitude regions;
(3) OI achieves more accurate estimates than interpolation and reanalysis estimates from (1) and (2), respectively, and
the complementary effect between reanalysis and interpolation estimates contributes to the large improvement of OI
estimates in sparsely gauged regions; and (4) ensemble precipitation estimates show good discrimination and
reliability performance for all thresholds, and the BSS values for ensemble precipitation and CRPSS values for
ensemble Tmean and Trange are high in most cases. BSS values of ensemble precipitation increase from 1979 to 2018
due to the increase of the number of stations.
Overall, EMDNA (version 1) will be useful for many applications in North America such as regional or continental
hydrological modeling. Meanwhile, we recognize that the current framework is not perfect and have provided
suggestions on the future directions for large-scale ensemble estimation of meteorological variables. Continuing
efforts from the community are needed to promote the development of probabilistic estimation methods and datasets.

**Author contributions:** GT and MC designed the framework of this study. GT collected data, performed the analyses
and wrote the paper. MC, SP, AN and AW contributed to the design of the methodology and result evaluation. SP,
DB and PW contributed to the evaluation of methodology and results. All authors contributed to data analysis,
discussions about the methods and results, and paper improvement.
**Competing interests:** The authors declare that they have no conflict of interest.
**Acknowledgment:** The study is funded by the Global Water Futures (GWF) program in Canada. The authors
appreciate the extensive efforts from the developers of the ground and reanalysis datasets to make their products
available. The authors also thank Federated Research Data Repository (FRDR; https://www.frdr-dfdr.ca; Access Date:
September 29, 2020) for publishing our dataset as open access to users.

**Appendix A. Regression coefficients**
The coefficients for locally weight linear regression are estimated using weighted least square. Given a station $i$ with
$m$ neighboring stations, let $\mathbf{A} = [1, A_1, \dots, A_n]$ be the $m \times n + 1$ attribute matrix, let $\mathbf{x} = (x_1, x_2, \dots, x_m)$ be the station
observations from neighboring stations, and let $\mathbf{w}_i = (w_{i,1}, w_{i,2}, \dots, w_{i,m})$ be the weight vector with distance-based



weights computed from Eq. (5). The regression coefficients $\boldsymbol{\beta} = (\beta_0, \beta_1, \dots, \beta_n)$ for Eq. (4) are estimated from the
weighted normal equation as

$$\boldsymbol{\beta} = (\mathbf{A}^{\mathrm{T}}\mathbf{W}\mathbf{A})^{-1}\mathbf{A}^{\mathrm{T}}\mathbf{W}\mathbf{x}, \qquad\qquad\qquad \text{A1}$$

where the $m \times m$ weight matrix $\mathbf{W} = \mathbf{I}_m \mathbf{w}_i$ is a diagonal matrix obtained by multiplying the $m \times m$ identity matrix
$\mathbf{I}_m$ with the weight vector $\mathbf{w}_i$.
The regression coefficients for logistic regression (Eq. 6) are estimated iteratively as:

$$\boldsymbol{\beta}^{new} = \boldsymbol{\beta}^{old} + (\mathbf{A}^{\mathrm{T}}\mathbf{W}\mathbf{V}\mathbf{A})^{-1}\mathbf{A}^{\mathrm{T}}\mathbf{W}(\mathbf{P}_0 - \boldsymbol{\pi}) \qquad\qquad \text{A2}$$

$$\boldsymbol{\pi} = \frac{1}{1 + \exp\left(-\mathbf{A}\boldsymbol{\beta}^{old}\right)} \qquad\qquad\qquad \text{A3}$$

$$\mathbf{V} = \mathbf{I}_m \boldsymbol{\pi}(1 - \boldsymbol{\pi}) \qquad\qquad\qquad \text{A4}$$

where $\mathbf{P}_0$ is a vector of binary precipitation occurrence for neighboring stations, $\boldsymbol{\pi}$ is the vector of estimated PoP for
neighboring stations, and $\mathbf{V}$ is the diagonal variance matrix for PoP. The regression coefficients $\boldsymbol{\beta}^{old}$ are initialized as
a vector of ones.

## Appendix B. Anomalous stations

To exclude climatologically anomalous stations, for temperature (Tmean or Trange), we calculate: (1) the absolute
difference of the climatological mean between the target station and the average value of its 10 neighboring stations
(referred as Diff-1), and (2) the absolute difference of the climatological mean between station observation and
regression estimates (referred as Diff-2). A temperature station will be excluded if its Diff-1 is larger than the 95%
percentile and its Diff-2 larger than the 99% percentile of all stations simultaneously. The threshold of percentiles for
Diff-1 is lower to better identify some climatologically anomalous stations.
For precipitation, the ratio (Ratio-1 and Ratio-2) is obtained in the same way with the Diff-1 and Diff-2 of temperature.
A two-tailed check is used for precipitation compared with the one-tailed check for temperature. A precipitation station
will be excluded if its Ratio-1 is larger (or smaller) than the 99.9% (1%) percentile and its Ratio-2 larger (or smaller)
than the 99.9% (1%) percentile simultaneously. This check has more tolerance for heavy precipitation but tries to
exclude more extremely dry stations.





As a result, ~1.5% precipitation and temperature stations are rejected, after which algorithms described in Sect. 3.1.1
and 3.1.2 are re-run. Stations can be anomalous because they are badly operated or simply because they are unique in
terms of topography or climate. The usage of Diff-2 or Ratio-2 is helpful to avoid excluding unique stations, but for
cases where the regression is ineffective, the unique stations can still be wrongly excluded. Although the effect on
final estimates could be rather small, better strategies could be used in future studies.

**Appendix C. Error of BMA-merged reanalysis estimates**
The errors of BMA-merged estimates are first estimated for all stations and then interpolated to grids. Considering
station observations cannot be used to evaluate merged estimates once they are used in bias correction or BMA weight
estimation, a two-layer cross-validation strategy is designed. In the first layer, we treat $i$ as the target station and find
its $m$ ($j_1 = 1, 2, \dots, m; \ i \notin j_1$) neighboring stations. In the second layer, we treat each $j_1$ as a target station, and (1)
find $m$ ($j_2 = 1, 2, \dots, m; \ i \notin j_2$) neighboring stations for each $j_1$, (2) calculate linear scaling correction factors for all
$j_2$, (3) estimate the correction factor for the target $j_1$ by interpolating factors at all $j_2$ stations using inverse distance
weighting, (4) correct estimates at $j_1$ using the correction factor, (5) calculate BMA weights of three reanalysis
products for all $j_1$ stations, (6) interpolate BMA weights from all $j_1$ stations to the target station $i$ and merge the three
reanalysis products for $i$, and (7) calculate the difference between merged reanalysis estimates and station observations
for $i$. This two-layer design may seem convoluted but is necessary to ensure that the error estimation is realistic. $j_1$
and $j_2$ could be partly overlapped due to their close locations but should not cause a large effect on the error estimation
for $i$ because data for $i$ are only used in (7) in this design. The station-based errors are interpolated to all grids using
inverse distance weighting.
**Appendix D. Metrics for probabilistic evaluation**
BSS is calculated based on the Brier Score (BS):

$$\text{BSS} = 1 - \frac{\text{BS}}{\text{BS}_{clim}} \qquad \text{D1}$$

$$\text{BS} = \frac{1}{n} \sum_{i=1}^{n} (\text{PoP}_{ens} - \text{PoP}_{obs})^2 \qquad \text{D2}$$

where $\text{PoP}_{ens}$ is the estimated probability of ensemble precipitation, $\text{PoP}_{obs}$ is the observed binary precipitation
occurrence, $n$ is the sample number, and $\text{BS}_{clim}$ is the climatological BS by assigning the climatological probability
to all samples. When the two series match the value of BSS will be equal to one.
CRPSS is calculated based on the continuous ranked probability skill score (CRPS; Hersbach, 2000):





$$CRPSS = 1 - \frac{CRPS}{CRPS_{clim}} \qquad\qquad D3$$

$$CRPS = \int_{-\infty}^{\infty} (F(x) - H(x \geq x_o))^2 dx \qquad\qquad D4$$

where $F(x)$ is the CDF of the ensemble temperature estimate $x$, $x_o$ is the observed temperature, $H(x \geq x_o)$ is the
Heaviside step function with the value being one if the condition $x \geq x_o$ is satisfied and zero if not satisfied, and
$CRPS_{clim}$ is the climatological CPRS. CRPS measures the distance between the CDF of probabilistic estimates and
observations. For a perfect match, the value of CRPSS would be one.
**Appendix E. Teaser dataset**
The teaser dataset is a subset of EMDNA probabilistic estimates for a small region (-116.8° to -115.2°W, 50.7° to
51.9°N) and a short period (2014 to 2015) with only 25 ensemble members. Users can easily download and preview
the teaser dataset (<30 MB) before downloading the entire EMDNA dataset (~3 TB or ~40 GB) as shown in Sect. 6.
The region covers the Bow River basin above Banff, Canada, which is located in the Canadian Rockies (Figure A1).
The spread of ensemble members in this region could be large due to the complex topography and limited stations.

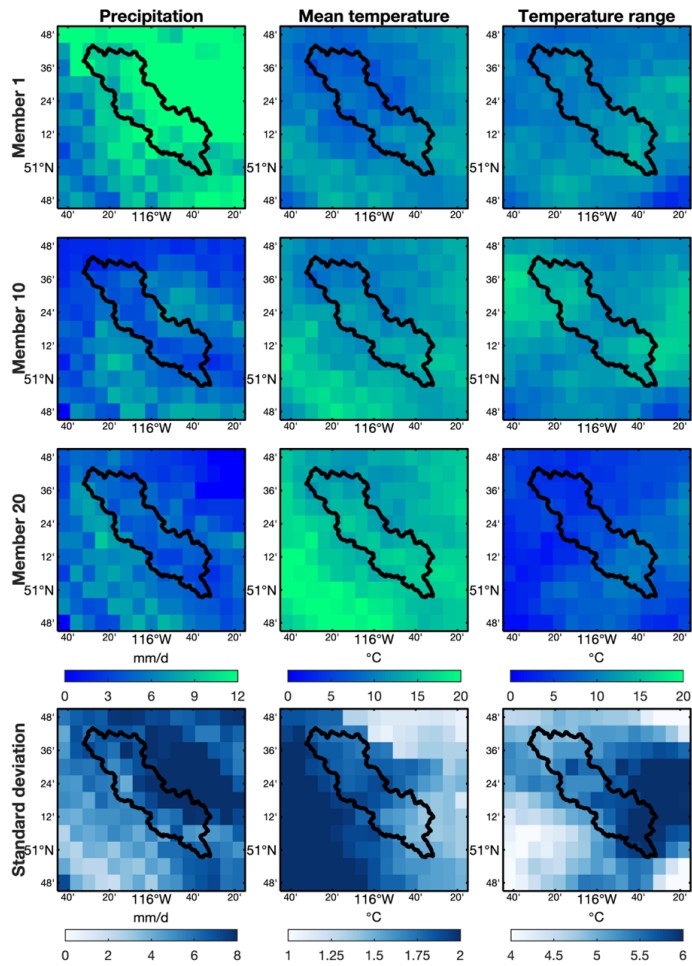


Figure A1. The distributions of daily precipitation (the first column), mean daily temperature (the second column), and daily temperature range (the third column) on 29 June 2015. The first to third rows represent ensemble members 1, 10, and 20, respectively. The fourth row represents the standard deviation of 25 members for this day. The black line outlines the Bow River basin above Banff, Canada.

700

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





Sevruk, B.: International comparison of national precipitation gauges with a reference pit gauge., WMO Instrum. Obs.
Methods Rep. No 17, 111, 1984.
Shen, Y., Zhao, P., Pan, Y. and Yu, J. J.: A high spatiotemporal gauge-satellite merged precipitation analysis over
China, J. Geophys. Res.-Atmospheres, 119(6), 3063–3075, doi:10.1002/2013jd020686, 2014a.
Shen, Y., Zhao, P., Pan, Y. and Yu, J.: A high spatiotemporal gauge-satellite merged precipitation analysis over China,
J. Geophys. Res. Atmospheres, 119(6), 3063–3075, doi:10.1002/2013JD020686, 2014b.
Shen, Y., Hong, Z., Pan, Y., Yu, J. and Maguire, L.: China's 1 km Merged Gauge, Radar and Satellite Experimental
Precipitation Dataset, Remote Sens., 10(2), 264, doi:10.3390/rs10020264, 2018.
Sinclair, S. and Pegram, G.: Combining radar and rain gauge rainfall estimates using conditional merging,
Atmospheric Sci. Lett., 6(1), 19–22, doi:10.1002/asl.85, 2005.
Slater, A. G. and Clark, M. P.: Snow Data Assimilation via an Ensemble Kalman Filter, J. Hydrometeorol., 7(3), 478–
493, doi:10.1175/JHM505.1, 2006.
Sun, Q., Miao, C., Duan, Q., Ashouri, H., Sorooshian, S. and Hsu, K.-L.: A Review of Global Precipitation Data Sets:
Data Sources, Estimation, and Intercomparisons, Rev. Geophys., doi:10.1002/2017rg000574, 2018.
Tang, G., Zeng, Z., Long, D., Guo, X., Yong, B., Zhang, W. and Hong, Y.: Statistical and Hydrological Comparisons
between TRMM and GPM Level-3 Products over a Midlatitude Basin: Is Day-1 IMERG a Good Successor for TMPA
3B42V7?, J. Hydrometeorol., 17(1), 121–137, doi:10.1175/jhm-d-15-0059.1, 2016.
Tang, G., Behrangi, A., Long, D., Li, C. and Hong, Y.: Accounting for spatiotemporal errors of gauges: A critical step
to evaluate gridded precipitation products, J. Hydrol., 559, 294–306, doi:10.1016/j.jhydrol.2018.02.057, 2018a.
Tang, G., Behrangi, A., Ma, Z., Long, D. and Hong, Y.: Downscaling of ERA-Interim Temperature in the Contiguous
United States and Its Implications for Rain–Snow Partitioning, J. Hydrometeorol., 19(7), 1215–1233,
doi:10.1175/jhm-d-18-0041.1, 2018b.
Tang, G., Clark, M. P., Papalexiou, S. M., Newman, A. J., Wood, A. W., Brunet, D. and Whitfield, P. H.: EMDNA:
Ensemble Meteorological Dataset for North America [Dataset], FRDR, doi:https://doi.org/10.20383/101.0275, 2020a.
Tang, G., Clark, M. P., Papalexiou, S. M., Ma, Z. and Hong, Y.: Have satellite precipitation products improved over
last two decades? A comprehensive comparison of GPM IMERG with nine satellite and reanalysis datasets, Remote
Sens. Environ., 240, 111697, doi:10.1016/j.rse.2020.111697, 2020b.
Tang, G., Clark, M. P., Newman, A. J., Wood, A. W., Papalexiou, S. M., Vionnet, V. and Whitfield, P. H.: SCDNA:
a serially complete precipitation and temperature dataset for North America from 1979 to 2018, Earth Syst. Sci. Data,
12(4), 2381–2409, doi:https://doi.org/10.5194/essd-12-2381-2020, 2020c.
Teutschbein, C. and Seibert, J.: Bias correction of regional climate model simulations for hydrological climate-change
impact studies: Review and evaluation of different methods, J. Hydrol., 456–457, 12–29,
doi:10.1016/j.jhydrol.2012.05.052, 2012.
Trenberth, K. E., Dai, A., Rasmussen, R. M. and Parsons, D. B.: The Changing Character of Precipitation, Bull. Am.
Meteorol. Soc., 84(9), 1205–1218, doi:10.1175/BAMS-84-9-1205, 2003.
Vila, D. A., de Goncalves, L. G. G., Toll, D. L. and Rozante, J. R.: Statistical Evaluation of Combined Daily Gauge
Observations and Rainfall Satellite Estimates over Continental South America, J. Hydrometeorol., 10(2), 533–543,
doi:10.1175/2008JHM1048.1, 2009.





Weedon, G. P., Balsamo, G., Bellouin, N., Gomes, S., Best, M. J. and Viterbo, P.: The WFDEI meteorological forcing
data set: WATCH Forcing Data methodology applied to ERA-Interim reanalysis data, Water Resour Res, 50(9), 7505–
7514, doi:10.1002/2014wr015638, 2014.
Willkofer, F., Schmid, F.-J., Komischke, H., Korck, J., Braun, M. and Ludwig, R.: The impact of bias correcting
regional climate model results on hydrological indicators for Bavarian catchments, J. Hydrol. Reg. Stud., 19, 25–41,
doi:10.1016/j.ejrh.2018.06.010, 2018.
Wood, A. W., Leung, L. R., Sridhar, V. and Lettenmaier, D. P.: Hydrologic Implications of Dynamical and Statistical
Approaches to Downscaling Climate Model Outputs, Clim. Change, 62(1), 189–216,
doi:10.1023/B:CLIM.0000013685.99609.9e, 2004.
Wu, H., Adler, R. F., Tian, Y., Huffman, G. J., Li, H. and Wang, J.: Real-time global flood estimation using satellite-
based precipitation and a coupled land surface and routing model, Water Resour. Res., 50(3), 2693–2717,
doi:10.1002/2013wr014710, 2014.
Xie, P. and Xiong, A.-Y.: A conceptual model for constructing high-resolution gauge-satellite merged precipitation
analyses, J. Geophys. Res. Atmospheres, 116(D21), doi:10.1029/2011JD016118, 2011.
Xu, S., Wu, C., Wang, L., Gonsamo, A., Shen, Y. and Niu, Z.: A new satellite-based monthly precipitation
downscaling algorithm with non-stationary relationship between precipitation and land surface characteristics, Remote
Sens. Environ., 162, 119–140, doi:10.1016/j.rse.2015.02.024, 2015.
Yamazaki, D., Ikeshima, D., Tawatari, R., Yamaguchi, T., O'Loughlin, F., Neal, J. C., Sampson, C. C., Kanae, S. and
Bates, P. D.: A high-accuracy map of global terrain elevations, Geophys. Res. Lett., 44(11), 5844–5853,
doi:10.1002/2017GL072874, 2017.
Yang, D., Goodison, B. E., Ishida, S. and Benson, C. S.: Adjustment of daily precipitation data at 10 climate stations
in Alaska: Application of World Meteorological Organization intercomparison results, Water Resour. Res., 34(2),
241–256, doi:10.1029/97WR02681, 1998.
Yang, D., Kane, D., Zhang, Z., Legates, D. and Goodison, B.: Bias corrections of long-term (1973-2004) daily
precipitation data over the northern regions, Geophys. Res. Lett., 32(19), n/a-n/a, doi:10.1029/2005gl024057, 2005.
Yin, J., Gentine, P., Zhou, S., Sullivan, S. C., Wang, R., Zhang, Y. and Guo, S.: Large increase in global storm runoff
extremes driven by climate and anthropogenic changes, Nat. Commun., 9(1), 4389, doi:10.1038/s41467-018-06765-
969    2, 2018.
