# Peer review of "EMDNA: Ensemble Meteorological Dataset for North America"

_Earth System Science Data, 2020_

## Referee Comment (RC1) · Graham Weedon (Referee) · 26 Mar 2021

Review of Tang et al. EMDNA: Ensemble meteorological dataset for North America.

One of the factors limiting the performance of land surface models (LSMs) and hydrological models (HMs) is the uncertainty in meteorological forcing data. There are a variety of ways his uncertainty can be assessed, but use of ensemble datasets that represent the plausible range of forcings is especially efficient/convenient. This paper provides both the methodology to produce ensemble meteorological datasets suitable for forcing LSMs and HMs and assesses a dataset already generated for North America, EMDNA, spanning 1979-2018 with a daily time step.

Overall, it has been a real pleasure to read such a well-written, clear and careful study. I strongly recommend acceptance with minor corrections. I list below a note about data accessibility, a few minor points and some small text corrections.

Data access:

Using http://doi.or/10.20383/101.0275 to access the EMDNA files via a Chrome browser the authors need to be aware that under each year a warning appears: "File list too long truncated." This means that the last file that can be seen is: EMDNA_YYYY.049.nc4. This may just be a browser-specific issue, but I suggest the authors recommend downloading via ftp and, if necessary, provide a protocol for this. Alternatively, it may be that there is a recommended browser that does not have this problem. Less important: Line 589 says: "The total data size is 3.35 TB.", but the site indicates 3.39 TB.

Minor points:

Line 107 "The possible dependence between reanalysis estimates and station data is not considered when merging them in this study." This is an important issue that has been under-played here. You should clarify for the reader whether surface observations (e.g. from SYNOPS) are actually incorporated into/constrain the various reanalyses. If so, you should spell out the effect this could have had on the results (i.e. the direction of bias).

Line 190 "leave-one-out cross-validation procedure" – perhaps mention that this is also known as the jackknife procedure.

Line 247 "orographic uplift": As I was originally trained as a geologist this sounds awfully like orogenesis or mountain building (which is not relevant at the time scale of this study). I suggest changing to e.g. "orographic uplift of air parcels/clouds".

Line 331 "in northern Canada and Alaska...where under-catch of precipitation is often large." Firstly, precipitation gauges can over-catch rain as well as under-catch so

that: "catch correction" or "gauge-catch correction" are more generic than "under-catch correction" (would need changing in several places). Secondly, I recommend that you explain that in Canada and Alaska the issue is the dominance of snowfall as a proportion of total precipitation (since snow catch errors are proportionately much larger issue than for rain). Thirdly you might want to add, to your notes that variances/errors are often higher in areas which are data sparse and topographically elevated, that the increased proportion of snowfall (and hence uncertainty due to catch-corrections) could be a correlated factor.

Lines 604-608 "four main steps" I suggest you include mention for catch-correction of precipitation via Beck et al's (2020) data.

Text corrections: Line 152: Reverse order of text: "gridded precipitation and temperature" > "temperature and gridded precipitation" [to agree with earlier mention of linear regression and logistic regression].

Equation 14: Should: $R_{t,PR} = rhoCRR_{t-1,TR} + \ldots$ be changed to $R_{t,PR} = rhoCRR_{t-1,PR} + \ldots$ [i.e TR > PR]?

Line 395: "The grid resolution is" > "For the improvement estimates the grid resolution is"

Line 427: "higher accuracy reanalysis estimates" > "higher accuracy than reanalysis estimates"

Line 523: "observation filed" > "observation field"

Line 527-528: "necessary considering OI" > "necessary considering that OI"

References: Check formatting because Karger et al., 2017, Mendoza et al., 2017 and Weedon et al., 2014 need full stops in the abbreviations of the journal names.

Graham P. Weedon

---

## Referee Comment (RC2) · Jonathan Gourley (Referee) · 6 Apr 2021

Reviewer Summary:

This article presents the methodology of the development of an historical meteorological dataset to be used for hydrologic modeling studies and beyond. The primary variables considered are daily precipitation, mean temperature, and diurnal temperature range. The uniqueness of the methodology is the development of 100 daily ensemble members as well as the optimal interpolation method based on 3 model reanalysis products used as background fields and station data used as observations. Overall, the presentation of the methodology is clear and understandable. I have a few comments that can be considered to improve the clarify of the presentation, but overall consider

this a novel and useful contribution to the field of hydrometeorology.

Major comments:

1. Considered variables - The meteorological dataset is comprised of 3 variables: mean daily temperature, mean daily temperature range, and daily precipitation amount. While these are common variables used in hydrologic modeling, it would be useful to the reader if the authors could provide some more justification on the use of these variables alone. The model reanalyses and station observations consider many more variables. If one were using a more complex Earth system model, for example, there could be a need for additional variables related to moisture, pressure, wind, and solar radiation. Consideration of the other standard variables would broaden the outreach of such a dataset.

2. Snow - The authors touch on the issue of gauge undercatch related to precipitation especially in mountainous regions and high latitudes. It's well known that the gauge efficiency with snow is much different than with rain. Were these treated the same or differently? Second, is it up to the user to determine if the "precipitation" is either in solid or liquid form? It would be useful to provide some more clear guidance on how to make this important hydrologic discrimination.

Minor comments:

1. Line 45: Change "reflectivity-rainfall relationships" to "representativeness of radar variables to surface rainfall." This more general as it considers dual-polarization variables (now used for rainfall estimation).

2. Line 48: Satellite data also have issues with data latency, which can limit their use in real-time applications.

3. Line 93-94: Change "northern to" to "north of" and "southern to" to "south of".

4. Line 110 (figure caption): Remove "radial".

5. Line 119-120: I am confusing Trange - as it can represent a grid cell with a large diurnal temperature range with little uncertainty or a grid cell with small diurnal temperature range but with large uncertainty bounds. These are quite different. Please clarify how this framework can be applied to a diurnal temperature range (comprised of two variables).

6. Line 316-319: But does this method apply separate corrections to rain and snow? Please specify.

7. Line 382-383: Why not use a normalized RMSE to remove the patterns caused by climatology? Also, was it not necessary to show the normalized bias because it was successfully removed?

8. Line 414-416: OK, now we're normalizing the variables by the mean as per my comment #7. But why just do it here and not before?

9. Line 435: I'd be OK if Fig. 8 were moved to an appendix or removed. It's pretty busy and didn't provide many insights.

10. Figure 9: Are ensemble members 1 and 100 randomly chosen or ranked? In either case, my eyes can distinguish no/little differences in the plots using the same color bar. Please explain and consider using a different color bar if you're trying to highlight differences.

11. Line 503-504: Regarding the lower CRPSS with Trange, is this because it's composed of both the daily max and min?

Jonathan J. Gourley

---

## Referee Comment (RC3) · Anonymous Referee #3 · 16 Apr 2021

Comments on the manuscript entitled "EMDNA: Ensemble Meteorological Dataset for North America"

General comments:

This study develops a new meteorological dataset EMDNA over North America. The dataset contains three variables: precipitation, mean daily temperature, and daily temperature range at a relatively high spatial resolution. Although there have been many other regional and global meteorological datasets, EMDNA stands out with the feature of probabilistic estimates, making it a useful choice for hydrometeorological studies such as the propagation of meteorological uncertainties to hydrological simulations. The dataset fits well the scope of the journal EESD. Overall, the manuscript is well

written and methods and analysis are clearly described, well discussion with existing products. I believe that this new dataset will a good and timely contribution to the hydrometeorology community.

I recommend accepting this manuscript with minor revision. Below are a few comments that could be useful to improve the manuscript.

Major comments:

1. The dataset provides both deterministic and probabilistic estimates. The ensemble mean is the average of 100 members, which is also provided in the final dataset. What's the difference between deterministic estimates from optimal interpolation-based merging and the ensemble mean from 100 members? For air temperature, the ensemble mean and optimal interpolation estimates should be similar. For precipitation, the methodology in this study uses transformed precipitation to calculate probabilistic estimates, and then applies back-transformation. Therefore, the ensemble mean may not be the same with the deterministic estimates. It would be useful to discuss their differences and make recommendations on what products should be used in practical applications.

2. The distance-weight function (Eq. 5) is cubic with the maximum distance defined based on the distance between target and neighboring stations. This means that the weight could be smooth in some cases compared to some sharp distance-weight functions such as the inverse distance weighting. It would be great if the authors can discuss the potential effect of the choice of functions?

3. It is useful to add more introduction to SCDNA because it is the most important data source of EMDNA and the independent validation of EMDNA also is related to SCDNA. In practice, many researchers often use raw station-based observations instead of gap filled datasets. More details on SCDNA would be nice to add here, so the readers do not have to read the original paper of the SCDNA.

Minor comments: 1. Line 98: It is better to mention why the three reanalysis products are used.

2. Line 151: The locally weighted linear regression is the same with the geographically weighted regression (GWR). It is better to clarify this concept here.

3. Line 190: The leave-one-out strategy may also be affected by the distributions of stations. For example, if two stations are very close, the leave-one-out may not be able to provide objective evaluation of them. Nevertheless, evaluation using independent stations also have a similar problem. The authors can add a little discussion here.

4. Line 320: Clarification of the streamflow sources will be useful to show whether the bias corrected climatology is reliable. I suppose there may be few streamflow stations in high latitude regions. Can the authors confirm by providing more details of the raw dataset?

5. Line 427: "than" should be added after "higher accuracy".

6. Line 568: GPM-IMERG needs a full name here: The Integrated Multi-satellitE Retrievals for GPM (IMERG). A proper citation is also needed. IMERG and weather radars cannot cover high latitude regions, and thus it is better to specify the scope in this sentence.

7. Line 569: It will be also useful to mention the error of temperature measurements, although such error could be small compared to precipitation undercatch.

---

## Referee Comment (RC4) · Daniel Wright (Referee) · 19 Apr 2021

The authors present an ensemble dataset of historical precipitation and temperature to support hydrologic analyses across North America. My review will be very brief, since three detailed reviews have already been obtained and they share a general concensus (which I share) that the manuscript is very strong and needs at most minor changes before publication.

I will admit that since there are already three reviews, I did not go as deep into the methods and results as I otherwise would have. So my first two comments may be things that are addressed in the methods. I have two general comments:

1. I understand that as a result of lower station densities at higher latitudes, the authors increase the search radius used to select stations for interpolation. While this is perfectly reasonable as a practical necessity, it isn't really right conceptually. Since the authors are blending in reanalysis products, an alternative way of thinking/dealing with this problem is that reanalysis should play a larger role in areas where station density is lowest. It may be that this is indeed how things shake out, but as I said, I didn't read things in enough detail to figure that out. Please comment.

2. My understanding of Newman's methods (e.g. Newman et al. 2015) from prior conversations is that station data are interpolated onto a grid of points that represent gridcell centers, and then the values at these center points are assumed to describe the grid-averaged value. I assume the same approach is taken in this study, albeit then modified with the reanalysis information. While I imagine this is a standard approach, I still have some concerns. First, and probably least, is that you are effectively representing grid-averaged precipitation based on station values. Second, due to the focus on the center of the grid cell, you could have a strange situation in areas with high station density, in which at least some members would say there is no precipitation in a grid cell, even if a gage within that cell (say near its edge) reports precipitation. How big an issue this could be in practice, I have no idea. Long story short: I'd be curious to see how the results of these sorts of methods differ when you interpolate them on a much finer grid (say 0.01 degrees) and then aggregate them back up to the final resolution (ie. 0.1 degrees). That would more or less address these conceptual problems.

3. While datasets such as this one are doubtless useful, anyone who knows me knows that I feel they have important limitations: specifically, I don't believe 0.1 degree daily is sufficient for realistic hydrologic simulations in many (perhaps most) landscapes. This is based on plenty of work by myself and others. The most directly relevant paper of mine being Sampson et al. (2020). I would urge the authors to ponder the more challenging issue of generating the subdaily, km-scale ensemble datasets that are needed for large-scale hydrologic predictions to actually work-if anyone has the skills to do it,

they do!

Reference: Sampson, Alexa A., Daniel B. Wright, Ryan D. Stewart, and Allison C. LoBue. "The Role of Rainfall Temporal and Spatial Averaging in Seasonal Simulations of the Terrestrial Water Balance." Hydrological Processes 34, no. 11 (May 30, 2020): 2531–42. https://doi.org/10.1002/hyp.13745.

Daniel Wright

---

## Author Comment (AC1) · 19 May 2021

**Response to comments**

The authors thank the editor David Carlson and four reviewers (Graham Weedon, Jonathan Gourley, Daniel Wright, and an anonymous reviewer) for their constructive comments on the manuscript and the development of the EMDNA dataset. We have carefully revised the manuscript and responded to all comments as shown in this response letter.

Guoqiang Tang on behalf of all co-authors

Centre for Hydrology, University of Saskatchewan

**Response to Graham Weedon (Referee 1):**

One of the factors limiting the performance of land surface models (LSMs) and hydrological models (HMs) is the uncertainty in meteorological forcing data. There are a variety of ways his uncertainty can be assessed, but use of ensemble datasets that represent the plausible range of forcings is especially efficient/convenient. This paper provides both the methodology to produce ensemble meteorological datasets suitable for forcing LSMs and HMs and assesses a dataset already generated for North America, EMDNA, spanning 1979-2018 with a daily time step.

Overall, it has been a real pleasure to read such a well-written, clear and careful study. I strongly recommend acceptance with minor corrections. I list below a note about data accessibility, a few minor points and some small text corrections.

Response: Thank you for the valuable comments. We have responded to your comments and revised the manuscript.

Data access:

Using http://doi.org/10.20383/101.0275 to access the EMDNA files via a Chrome browser the authors need to be aware that under each year a warning appears: "File list too long truncated." This means that the last file that can be seen is: EMDNA_YYYY.049.nc4. This may just be a browser-specific issue, but I suggest the authors recommend downloading via ftp and, if necessary, provide a protocol for this. Alternatively, it may be that there is a recommended browser that does not have this problem. Less important: Line 589 says: "The total data size is 3.35 TB.", but the site indicates 3.39 TB.

Response: We found the same problem of "File list too long truncated" using Chrome and other browsers, which is due to the display limitation of the FRDR website. Users can view and download the entire dataset using Globus (see the "Download Dataset" button on the dataset

webpage which also navigates to a tutorial video of Globus). Globus is a powerful tool for transferring all types of datasets and is more stable than direct browser downloading. We understand that many researchers, including the authors, often use ftp to transfer and download datasets. However, FRDR does not provide ftp downloading. We released EMDNA on FRDR because of its acceptance of very large datasets and the connection between our project and the FRDR site.

The description was a little misleading in the original manuscript. The probabilistic dataset is 3.35 TB, and the deterministic dataset is 40.84 GB. The total size of 3.39 TB is shown on the webpage. We have revised the description in Section "6. Data availability"

"The data sizes are 3.35 TB for the probabilistic part and 40.84 GB for the deterministic part, respectively."

Minor points:

Line 107 "The possible dependence between reanalysis estimates and station data is not considered when merging them in this study." This is an important issue that has been under-played here. You should clarify for the reader whether surface observations (e.g. from SYNOPS) are actually incorporated into/constrain the various reanalyses. If so, you should spell out the effect this could have had on the results (i.e. the direction of bias).

Response: The three reanalysis products do not provide lists of stations used in their production. Whilst some surface observations are used by reanalysis products, none of the reanalysis products assimilate precipitation observations from station data. In many practical applications, the dependence between reanalysis model estimates and station data is often not considered, such as the MSWEP dataset which merges multiple station, satellite and reanalysis datasets (Beck et al., 2019). This is possibly because reanalysis estimates show large uncertainties according to ground observation-based evaluation studies. For EMDNA, the three reanalysis products (ERA5, JRA-55, and MERRA2) use different data assimilation systems and data sources, making it even harder to predict the impact of individual data sources. Nevertheless, the possible dependence between reanalysis datasets and stations could affect the performance of data merging because the calculation of station-based weights may not be perfect. We added some discussion in the revised manuscript.

"The dependence of reanalysis estimates on station data may have a negative effect on the merging of reanalysis products (Section 3.2) because the reanalysis dataset which assimilates more station data could be given higher weight. The potential dependence, however, is not considered in this study because of the limited understanding of the dependence between reanalysis estimates and station observations. Moreover, none of the reanalysis datasets assimilate precipitation data from stations."

Beck, H. E., E. F. Wood, M. Pan, C. K. Fisher, D. G. Miralles, A. I. J. M. van Dijk, T. R. McVicar, and R. F. Adler, 2019: MSWEP V2 Global 3-Hourly 0.1° Precipitation: Methodology and Quantitative Assessment. Bull. Amer. Meteor. Soc., 100, 473–500, https://doi.org/10.1175/BAMS-D-17-0138.1.

Line 190 "leave-one-out cross-validation procedure" – perhaps mention that this is also known as the jackknife procedure.

Response: The leave-one-out validation method is not actually the same as the jackknife method. The jackknife method calculates the uncertainty in an estimate by successively removing observations and calculating the standard error from the set of leave-one-out samples. Cross-validation compares predictions to each withheld sample.

Line 247 "orographic uplift": As I was originally trained as a geologist this sounds awfully like orogenesis or mountain building (which is not relevant at the time scale of this study). I suggest changing to e.g. "orographic uplift of air parcels/clouds".

Response: This description has been deleted in the revised manuscript. We will pay attention to this issue in future work.

Line 331 "in northern Canada and Alaska. . .where under-catch of precipitation is often large." Firstly, precipitation gauges can over-catch rain as well as under-catch so that: "catch correction" or "gauge-catch correction" are more generic than "under-catch correction" (would need changing in several places). Secondly, I recommend that you explain that in Canada and Alaska the issue is the dominance of snowfall as a proportion of total precipitation (since snow catch errors are proportionately much larger issue than for rain). Thirdly you might want to add, to your notes that variances/errors are often higher in areas which are data sparse and topographically elevated, that the increased proportion of snowfall (and hence uncertainty due to catch-corrections) could be a correlated factor.

Response: We have added discussion about gauge over-catch. We did not change "under-catch correction" terms to the more generic "gauge-catch correction" because the PBCOR dataset, which is used as correction reference in this study, truncates the correction ratio at one to ensure that adjusted precipitation cannot be smaller than raw precipitation (Beck et al., 2020). This means the design of PBCOR does not consider the over-catch problem, and thus the correction based on PBCOR in this study cannot overcome this limitation. Besides, we have revised the manuscript according to your second and third suggestions.

"Note that the rain gauge catch error includes both under-catch and over-catch. The potential over-catch could be caused by splash of rain or blow snow collected on the wind shield (Folland, 1988; Zhang et al., 2019). Since over-catch is less common compared to under-catch and the PBCOR dataset does not consider over-catch, the bias correction in this study only addresses the under-catch problem."

"The bias correction notably increases precipitation in northern Canada and Alaska (Fig. 2d) where the precipitation under-catch is often significant due to the large proportion of snowfall. The uncertainties of gridded estimates are typically larger in high-latitude sparsely gauged regions and topographically elevated regions, which is partly related to the increased proportion of snowfall and hence larger gauge catch errors."

Beck, H. E., E. F. Wood, T. R. McVicar, M. Zambrano-Bigiarini, C. Alvarez-Garreton, O. M. Baez-Villanueva, J. Sheffield, and D. N. Karger, 2020: Bias Correction of Global High-Resolution Precipitation Climatologies Using Streamflow Observations from 9372 Catchments. J. Climate, 33, 1299–1315, https://doi.org/10.1175/JCLI-D-19-0332.1.

Lines 604-608 "four main steps" I suggest you include mention for catch-correction of precipitation via Beck et al's (2020) data.

Response: We have added the PBCOR dataset.

Text corrections: Line 152: Reverse order of text: "gridded precipitation and temperature" > "temperature and gridded precipitation" [to agree with earlier mention of linear regression and logistic regression].

Response: We have changed the expression.

Equation 14: Should: $R_{t,PR} = \rho_{CRR} R_{t-1,TR} + . . .$ be changed to $R_{t,PR} = \rho_{CRR} R_{t1,PR} + . . .$ [i.e TR > PR]?

Response: Thank you for pointing out this mistake. We have corrected it.

Line 395: "The grid resolution is" > "For the improvement estimates the grid resolution is"

Response: We have modified the sentence in a different way which still agrees with your suggestion.

"The maps are at the 0.5° resolution, and the value of each 0.5° grid point is the median metric of all stations located within the grid."

Line 427: "higher accuracy reanalysis estimates" > "higher accuracy than reanalysis estimates"

Response: We have revised this sentence.

Line 523: "observation filed" > "observation field"

Response: We corrected this typing error.

Line 527-528: "necessary considering OI" > "necessary considering that OI"

Response: We have revised this sentence.

References: Check formatting because Karger et al., 2017, Mendoza et al., 2017 and Weedon et al., 2014 need full stops in the abbreviations of the journal names.

Response: We have corrected the problems in the references.

**Response to Jonathan Gourley (Referee 2):**

Reviewer Summary:

This article presents the methodology of the development of an historical meteorological dataset to be used for hydrologic modeling studies and beyond. The primary variables considered are daily precipitation, mean temperature, and diurnal temperature range. The uniqueness of the methodology is the development of 100 daily ensemble members as well as the optimal interpolation method based on 3 model reanalysis products used as background fields and station data used as observations. Overall, the presentation of the methodology is clear and understandable. I have a few comments that can be considered to improve the clarify of the presentation, but overall consider this a novel and useful contribution to the field of hydrometeorology.

Response: Thank you for the valuable comments. We have responded to your comments and revised the manuscript as follows.

Major comments:

1. Considered variables - The meteorological dataset is comprised of 3 variables: mean daily temperature, mean daily temperature range, and daily precipitation amount. While these are common variables used in hydrologic modeling, it would be useful to the reader if the authors could provide some more justification on the use of these variables alone. The model reanalyses and station observations consider many more variables. If one were using a more complex Earth system model, for example, there could be a need for additional variables related to moisture, pressure, wind, and solar radiation. Consideration of the other standard variables would broaden the outreach of such a dataset.

Response: We chose the three variables because they are the most common meteorological variables measured by ground stations and used in hydrometeorological applications. Other variables such as wind speed and humidity are only measured by a small number of stations. Therefore, the development of a continental dataset for those variables is challenging. The authors do plan to develop datasets with more variables to meet the requirement of complex models by collecting more data and developing new methods, which, however, is beyond the scope of EMDNA. We have added the explanation in the revised manuscript about why we choose the three variables.

"We select precipitation and temperature because they are used in many hydrometeorological studies and are measured by a large number of meteorological stations, while other variables (e.g., humidity and wind speed) are only measured by a much smaller collection of stations."

2. Snow - The authors touch on the issue of gauge undercatch related to precipitation especially in mountainous regions and high latitudes. It's well known that the gauge efficiency with snow is much different than with rain. Were these treated the same or differently? Second, is it up to the user to determine if the "precipitation" is either in solid or liquid form? It would be useful to provide some more clear guidance on how to make this important hydrologic discrimination.

Response: Snow is an important issue. We did not distinguish between rain and snow in this study. The under-catch correction method uses monthly maps of precipitation climatology, which implicitly considers the seasonal variation of rain and snow but cannot explicitly separate rain and snow events. The is a limitation of the current method because the under-catch is more severe for snowfall than rainfall as you pointed out.

The dataset does not provide precipitation phase information and users have to determine phase by themselves. We added two possible and simple options in the revised manuscript: (1) temperature threshold-based rain-snow classification which can be realized based on EMDNA temperature estimates; and (2) model-based classification which can utilize snow/rain outputs from reanalysis models to calculate snowfall ratio.

Station-based under-catch correction needs precipitation phase information and thus can provide rain/snow outputs. However, we don't use the station-based method as discussed in Section "5.4 Precipitation under-catch". Relevant discussion has been added in the manuscript.

"In addition, this correction method aims to constrain the total precipitation amount and cannot distinguish between rainfall and snowfall which show different gauge catch performance. Data users can realize rain-snow classification using approaches such as temperature threshold-based methods and reanalysis model-based snowfall proportion."

Minor comments:

1. Line 45: Change "reflectivity-rainfall relationships" to "representativeness of radar variables to surface rainfall." This more general as it considers dual-polarization variables (now used for rainfall estimation).

Response: We have changed the expression.

2. Line 48: Satellite data also have issues with data latency, which can limit their use in real-time applications.

Response: We have added data latency in this sentence.

3. Line 93-94: Change "northern to" to "north of" and "southern to" to "south of".

Response: We have made the change.

4. Line 110 (figure caption): Remove "radial".

Response: We have removed "radial".

5. Line 119-120: I am confusing Trange - as it can represent a grid cell with a large diurnal temperature range with little uncertainty or a grid cell with small diurnal temperature range but with large uncertainty bounds. These are quite different. Please clarify how this framework can be applied to a diurnal temperature range (comprised of two variables).

Response: Tmean and Trange are both inferred from two variables (minimum and maximum daily temperature). In the production of the dataset, Trange is treated as an independent variable. The gridded values and uncertainties are estimated and used to define the normal distribution. The probabilistic Trange is estimated by sampling from the normal distribution. The relationship

between the magnitudes of the diurnal temperature range and uncertainty won't affect the estimation procedure.

6. Line 316-319: But does this method apply separate corrections to rain and snow? Please specify.

Response: No, rain and snow are not separated by this method because the reference dataset only provides total precipitation amounts. We added clarification in the revised manuscript.

7. Line 382-383: Why not use a normalized RMSE to remove the patterns caused by climatology? Also, was it not necessary to show the normalized bias because it was successfully removed?

Response: We have changed RMSE to NRMSE to be consistent with Section 4.2, which is also mentioned in the next comment. The correction only removes the climatology bias and thus mean error or mean bias is not necessary. Nevertheless, daily-scale NRMSE or RMSE is still useful because the correction cannot remove the daily difference.

8. Line 414-416: OK, now we're normalizing the variables by the mean as per my comment #7. But why just do it here and not before?

Response: Please see our response to the previous comment.

9. Line 435: I'd be OK if Fig. 8 were moved to an appendix or removed. It's pretty busy and didn't provide many insights.

Response: We have deleted Fig. 8 and relevant texts to keep the manuscript more concise.

10. Figure 9: Are ensemble members 1 and 100 randomly chosen or ranked? In either case, my eyes can distinguish no/little differences in the plots using the same color bar. Please explain and consider using a different color bar if you're trying to highlight differences.

Response: Yes, they are randomly chosen. The differences are more notable for precipitation than for temperature. This because temperature estimates have much smaller uncertainties. Besides, it is not expected the different members show a very large difference on the monthly scale. The figure for the teaser dataset (Figure A1) shows a more obvious difference between members at the daily scale. We tested other color bars such as the rainbow style as shown below. The visualization effect is still similar to Figure 9 (i.e., notable difference for precipitation but small difference for temperature). Therefore, we do not change the raw Figure 9.

[Figure]

Figure: Same with Figure 9 in the main text but with a different color bar.

11. Line 503-504: Regarding the lower CRPSS with Trange, is this because it's composed of both the daily max and min?

Figure: This could be a major reason although Tmean is also composed of both daily max and min. Trange has worse deterministic estimates than Tmean probably because the bias direction (i.e., overestimation or underestimation) of Tmin and Tmax could be different resulting in the larger bias of Trange than Tmean. It is easier to capture the average status (i.e., Tmean) than the variation range (i.e., Trange). The larger bias of Trange deterministic estimates results in the lower CRPSS of Trange probabilistic estimates. We added discussion in this sentence.

"Trange shows lower CRPSS probably because the bias direction (i.e., overestimation or underestimation) of daily minimum and maximum temperature could be different, resulting in the larger bias of Trange than Tmean."

**Response to Referee 3:**

General comments:

This study develops a new meteorological dataset EMDNA over North America. The dataset contains three variables: precipitation, mean daily temperature, and daily temperature range at a relatively high spatial resolution. Although there have been many other regional and global meteorological datasets, EMDNA stands out with the feature of probabilistic estimates, making it a useful choice for hydrometeorological studies such as the propagation of meteorological uncertainties to hydrological simulations. The dataset fits well the scope of the journal EESD. Overall, the manuscript is well written and methods and analysis are clearly described, well discussion with existing products. I believe that this new dataset will a good and timely contribution to the hydrometeorology community. I recommend accepting this manuscript with minor revision. Below are a few comments that could be useful to improve the manuscript.

Response: Thank you for the valuable comments. We have responded to your comments and revised the manuscript as below.

Major comments:

1. The dataset provides both deterministic and probabilistic estimates. The ensemble mean is the average of 100 members, which is also provided in the final dataset. What's the difference between deterministic estimates from optimal interpolation-based merging and the ensemble mean from 100 members? For air temperature, the ensemble mean and optimal interpolation estimates should be similar. For precipitation, the methodology in this study uses transformed precipitation to calculate probabilistic estimates, and then applies back-transformation. Therefore, the ensemble mean may not be the same with the deterministic estimates. It would be useful to discuss their differences and make recommendations on what products should be used in practical applications.

Response: We added explanations in Section "6. Data availability". As you pointed out, the ensemble mean of Tmean and Trange is very similar to deterministic estimates because of the feature of the normal distribution. For precipitation, as a whole, the ensemble mean is slightly higher than deterministic estimates because of the back transformation process. We recommend using the deterministic estimates rather than the ensemble mean if data users do not need to quantify uncertainties in their applications.

"The ensemble mean of the 100 members for Tmean and Trange is similar to deterministic OI estimates. For precipitation, the ensemble mean is slightly higher than deterministic OI estimates due to the back transformation. We recommend that users select the deterministic dataset instead of the ensemble mean if their applications do not involve uncertainty characterization."

2. The distance-weight function (Eq. 5) is cubic with the maximum distance defined based on the distance between target and neighboring stations. This means that the weight could be smooth in some cases compared to some sharp distance-weight functions such as the inverse distance weighting. It would be great if the authors can discuss the potential effect of the choice of functions?

Response: Yes, the cubic weight function could be smoother in some cases. This means that the degradation rate of weights with increasing distance could be smaller than some other weight functions, resulting in smoother spatial distributions. We have added more discussion in Section "3.1.1 Locally weighted linear regression" where the weight function is introduced.

"The cubic weight function is smoother compared to functions such as exponential functions and inverse distance functions, indicating that $w_{ij}$ degrades with distance in a relatively slow way which generally leads to smooth spatial variations of variables. The comparison of different weight functions could be a future study direction."

3. It is useful to add more introduction to SCDNA because it is the most important data source of EMDNA and the independent validation of EMDNA also is related to SCDNA. In practice, many researchers often use raw station-based observations instead of gap filled datasets. More details on SCDNA would be nice to add here, so the readers do not have to read the original paper of the SCDNA.

Response: Yes, we agree that more details about SCDNA can increase the readability. We have added more details in Section "2. Datasets", including the nine steps used to produce SCDNA.

Minor comments:

1. Line 98: It is better to mention why the three reanalysis products are used.

Response: We have added explanation in Section "2. Datasets". The three products are very widely used. We used them because (1) they have relatively high spatiotemporal resolutions compared to other reanalysis products, and (2) their temporal length generally agrees with our target period (1979-2018).

2. Line 151: The locally weighted linear regression is the same with the geographically weighted regression (GWR). It is better to clarify this concept here.

Response: We have mentioned GWR in this sentence.

3. Line 190: The leave-one-out strategy may also be affected by the distributions of stations. For example, if two stations are very close, the leave-one-out may not be able to provide objective evaluation of them. Nevertheless, evaluation using independent stations also have a similar problem. The authors can add a little discussion here.

Response: We have added some discussion in Section "3.1.2 Locally weighted logistic regression".

"The leave-one-out evaluation could be affected by the distributions of stations in some cases. For example, two stations with very close distance may both show very high accuracy in the leave-one-out evaluation (this is a problem for all station-based evaluation methods)."

4. Line 320: Clarification of the streamflow sources will be useful to show whether the bias corrected climatology is reliable. I suppose there may be few streamflow stations in high latitude regions. Can the authors confirm by providing more details of the raw dataset?

Response: We have added more details about the streamflow sources.

"… streamflow observations collected from seven national and international sources, among which the Global Runoff Data Centre (GRDC), the U.S. Geological Survey (USGS), and the Water Survey of Canada Hydrometric Data (HYDAT) are data sources in North America. The streamflow stations are scarce in high latitude regions and absent in Greenland (Beck et al., 2020)."

5. Line 427: "than" should be added after "higher accuracy".

Response: We have added "than".

6. Line 568: GPM-IMERG needs a full name here: The Integrated Multi-satellitE Retrievals for GPM (IMERG). A proper citation is also needed. IMERG and weather radars cannot cover high latitude regions, and thus it is better to specify the scope in this sentence.

Response: We have deleted GPM-IMERG from this sentence because we refer to broad satellite precipitation sources and it is not essential to mention GPM-IMERG here. Deleting GPM-IMERG can make this sentence more concise. We will pay attention to the full name and citation in future papers.

We have specified the scope of satellites and radars in this sentence. Since their coverage is not the same, the statement is a little ambiguous.

"Moreover, including other data sources such as satellite and weather radar estimates is also an opportunity for regions with adequate sample coverage."

7. Line 569: It will be also useful to mention the error of temperature measurements, although such error could be small compared to precipitation undercatch.

Response: We have added discussion on the error of temperature measurement in Section "5.4 Precipitation under-catch".

"Station temperature measurements also contain errors due to microclimate and sensor design (Lin and Hubbard, 2004), which is generally small and not discussed here."

**Response to Daniel Wright (Referee 4):**

General comments:

The authors present an ensemble dataset of historical precipitation and temperature to support hydrologic analyses across North America. My review will be very brief, since three detailed reviews have already been obtained and they share a general concensus (which I share) that the manuscript is very strong and needs at most minor changes before publication.

I will admit that since there are already three reviews, I did not go as deep into the methods and results as I otherwise would have. So my first two comments may be things that are addressed in the methods. I have two general comments:

Response: Thank you for the valuable comments. We have responded to your comments and revised the manuscript.

1. I understand that as a result of lower station densities at higher latitudes, the authors increase the search radius used to select stations for interpolation. While this is perfectly reasonable as a practical necessity, it isn't really right conceptually. Since the authors are blending in reanalysis products, an alternative way of thinking/dealing with this problem is that reanalysis should play a larger role in areas where station density is lowest. It may be that this is indeed how things shake out, but as I said, I didn't read things in enough detail to figure that out. Please comment.

Response: The reason that we merge station data and reanalysis estimates in this study is to utilize the advantages of reanalysis products in areas with fewer stations to some extent. We use the

optimal interpolation (OI) to merge station and reanalysis data. The mains steps are briefly described here. (1) Locally weighted linear/logistic regression is used to obtain estimates corresponding to station locations using the leave-one-out strategy. The difference between regression estimates and station observations is treated as regression errors. Note this step obtains estimates corresponding to station locations instead of gridded estimates (Section 3.1). (2) We match reanalysis estimates and station data. The difference between reanalysis and stations is treated as reanalysis errors and is interpolated to the 0.1-degree grids (Section 3.2). (3) The OI is used to merge station regression estimates and reanalysis estimates based on weights calculated from regression errors and reanalysis errors (Section 3.3). The advantages of this design are (1) weights and inputs closely match each other and (2) weights in sparsely gauged regions are not determined by parameters fitted in densely gauged regions. In regions where the station density is lower, regression errors could be larger, resulting in the lower contribution of regression estimates and larger contribution of reanalysis estimates. A typical example is Greenland, where the distributions of OI estimates are close to reanalysis estimates due to the limited coverage of stations. Another possibility is that reanalysis estimates show large errors in sparsely gauged regions due to the poor model performance and limited data assimilation, and thus may not overwhelm regression estimates. In summary, the error-based weighting scheme has the advantage in determining the optimal combination of reanalysis and regression estimates. We have added more discussion about the station density in Section "3.1.1 Optimal Interpolation".

"In regions with few stations, the errors of regression estimates could be larger than reanalysis estimates, resulting in a smaller contribution from regression estimates and a larger contribution from reanalysis estimates, which is the complementary effect we expect by involving reanalysis datasets in EMDNA."

2. My understanding of Newman's methods (e.g. Newman et al. 2015) from prior conversations is that station data are interpolated onto a grid of points that represent gridcell centers, and then the values at these center points are assumed to describe the grid-averaged value. I assume the same approach is taken in this study, albeit then modified with the reanalysis information. While I imagine this is a standard approach, I still have some concerns. First, and probably least, is that you are effectively representing grid-averaged precipitation based on station values. Second, due to the focus on the center of the grid cell, you could have a strange situation in areas with high station density, in which at least some members would say there is no precipitation in a grid cell, even if a gage within that cell (say near its edge) reports precipitation. How big an issue this could be in practice, I have no idea. Long story short: I'd be curious to see how the results of these sorts of methods differ when you interpolate them on a much finer grid (say 0.01 degrees) and then aggregate them back up to the final resolution (ie. 0.1 degrees). That would more or less address these conceptual problems.

Response: We used Newman's locally weighted linear/logistic regression methods but implemented the methods in a different way. We obtained regression estimates at station locations using the leave-one-out strategy (see our response to your first comment). The problem of using the grid center to represent the grid average is avoided. Aggregating 0.01-degree estimates to 0.1-degree estimates could obtain authentic grid average estimates. The potential challenges of implementing this aggregation in a large domain are the computation cost and storage space. We may test this idea in future work.

3. While datasets such as this one are doubtless useful, anyone who knows me knows that I feel they have important limitations: specifically, I don't believe 0.1 degree daily is sufficient for realistic hydrologic simulations in many (perhaps most) landscapes. This is based on plenty of work by myself and others. The most directly relevant paper of mine being Sampson et al. (2020). I would urge the authors to ponder the more challenging issue of generating the subdaily, km-scale ensemble datasets that are needed for large-scale hydrologic predictions to actually work-if anyone has the skills to do it, they do!

Reference: Sampson, Alexa A., Daniel B. Wright, Ryan D. Stewart, and Allison C. LoBue. "The Role of Rainfall Temporal and Spatial Averaging in Seasonal Simulations of the Terrestrial Water Balance." Hydrological Processes 34, no. 11 (May 30, 2020): 2531–42. https://doi.org/10.1002/hyp.13745.

Response: Thank you for the suggestion. We do have the plan to develop datasets with higher resolutions to meet the requirement of broader hydrometeorological applications, while this may not be easy concerning both methodology and software/hardware. For example, the daily 0.1-degree dataset from 1979 to 2018 has a size of ~3.4 TB. Generating and releasing an hourly km-scale ensemble dataset will need enormous storage space. We have added discussion about the sub-daily and km-scale datasets in Section "7. Summary and Conclusions".

"Development of datasets at higher resolutions (e.g., 1 km and hourly) is also an important direction to enable more sophisticated hydrometeorological studies (e.g., Sampson et al., 2020)."